# GeoGramBench: Benchmarking the Geometric Program Reasoning in Modern LLMs

**Shixian Luo**[*†]**, Zezhou Zhu**[*†]**, Yu Yuan**[†]**, Yuncheng Yang**[†]**, Lianlei Shan**[†]**, Yong Wu**[‡†]
Li Auto Inc.[†]
`wuyong5@lixiang.com`

## Abstract

Geometric spatial reasoning forms the foundation of many applications in artificial intelligence, yet the ability of large language models (LLMs) to operate over geometric spatial information expressed in procedural code remains underexplored. In this paper, we address this gap by formalizing the `Program-to-Geometry` task, which challenges models to translate programmatic drawing code into accurate and abstract geometric reasoning. To evaluate this capability, we present **GeoGramBench**, a benchmark of 500 carefully refined problems organized by a tailored three-level taxonomy that considers geometric complexity rather than traditional mathematical reasoning complexity. Our comprehensive evaluation of 19 frontier LLMs reveals consistent and pronounced deficiencies: even the most advanced models achieve less than 50% accuracy at the highest abstraction level. By systematically analyzing model behaviors, our study exposes key limitations in program-driven spatial reasoning and positions GeoGramBench as an important resource for benchmarking and advancing behavioral research in symbolic-to-spatial geometric reasoning. Project page: https://github.com/LiAuto-DSR/GeoGramBench

## 1 Introduction

Spatial reasoning is fundamental to both human cognition and artificial intelligence, supporting applications ranging from robotics and autonomous navigation to automated design (Davis et al., 2011). With the rise of large language models (LLMs), interest has grown in evaluating their ability to interpret geometric transformations and spatial relations in complex environments (Yang et al., 2024b; Tang et al., 2025).

Mathematical geometric spatial reasoning is a specialized subdomain of spatial reasoning, requiring models to comprehend intricate geometric relationships and perform deep spatial reasoning. Researchers have recently developed multiple benchmarks including Mathverse (Zhang et al., 2024b), GeoSense (Xu et al., 2025), and Euclid (Zhang et al., 2024a) to assess LLMs' capabilities in visual geometry comprehension. Another emerging direction leverages procedural geometric code, such as Asymptote code, as a symbolic and structured interface for expressing geometry problems and probing spatial reasoning. While some existing benchmarks (e.g., AIME24 (MAA, 2025), MATH-500 (Zhang et al., 2024b)) include subsets containing Asymptote code, there is a lack of systematic, dedicated benchmarks specifically designed to evaluate LLMs' ability to perform program-driven spatial geometric reasoning. In this work, we formalize this unique setting as the `Program-to-Geometry` task, referring to the translation and abstraction process from procedural code to internal spatial representations.

Preliminary studies (Muennighoff et al., 2025) have shown that current LLMs struggle to bridge procedural geometry code to spatial reasoning. We expanded these investigations on a broader range of models further corroborate these observations, confirming this pronounced deficiency. For example, as shown in Figure 1, advanced models such as DeepSeek-R1 (Guo et al., 2025) suffer substantial drops in accuracy: 23.5% in AIME24 and 10.9% in MATH-500, when transitioning from text-only problems

---

[*]Equal contribution.
[‡]Corresponding author.

($\mathbb{P}_T$) to those with embedded procedural code ($\mathbb{P}_{TC}$). Similar trends are observed for models such as GPT-o1 (Jaech et al., 2024) and QwQ-32B (Team, 2025a), collectively indicating critical limitations in their ability to construct reliable spatial representations from symbolic code. Furthermore, recent work (Albalak et al., 2025) has highlighted the need to explore `Program-to-Geometry` spatial abstraction as a promising and under-investigated research direction.

Motivated by these findings, we introduce **GeoGramBench**, a dataset of 500 curated problems incorporating programmatic drawing code, designed to systematically assess both spatial-geometric abstraction capabilities and mathematical reasoning in LLMs. Our proposed taxonomy organizes problems into three categories: *Primitive Recognition*, *Local Relation Composition*, and *Global Abstract Integration*, based on the geometric complexity encoded in procedural code rather than traditional reasoning difficulty. Evaluation of 19 frontier LLMs reveals that even models such as the reasoning-oriented GPT-o1 achieve less than 50% accuracy on the most challenging level, underscoring the unique difficulty of this task and the urgent need for advances in spatial-reasoning model design.

This work makes the following contributions:

- We formalize the `Program-to-Geometry` translation task as a critical and underexplored capability for LLMs, encompassing not only the interpretation of procedural drawing code but also the downstream geometric reasoning it enables.

- We present **GeoGramBench**, a rigorously curated benchmark of 500 geometry problems with explicit procedural code, organized by a three-level taxonomy that enables comprehensive and fine-grained assessment of `Program-to-Geometry` competence.

- We conduct an extensive evaluation of 19 models, providing accuracy metrics and detailed behavior analyses aligned with our research questions. Our results highlight persistent weaknesses in geometric program reasoning, establishing GeoGramBench as a novel evaluation axis and fostering future advancements in spatially-grounded, symbolically-rich model training and analysis.

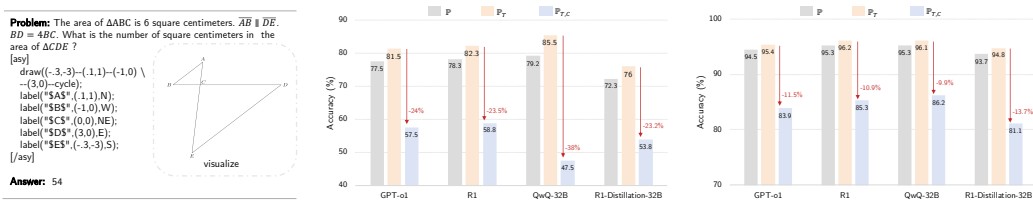

(a) Example of a problem from $\mathbb{P}_{TC}$ in MATH-500.

(b) Accuracy comparison of models on $\mathbb{P}_T$ vs. $\mathbb{P}_{TC}$ in AIME24.

(c) Accuracy comparison of models on $\mathbb{P}_T$ vs. $\mathbb{P}_{TC}$ in MATH-500.

Figure 1: Overview and performance analysis on text-only ($\mathbb{P}_T$) and text+code ($\mathbb{P}_{TC}$) geometry problems. (a) The procedural code is wrapped with `[asy][/asy]` and its geometric figure is visualized to facilitate understanding. (b) and (c) show accuracy comparisons of models on $\mathbb{P}_T$ and $\mathbb{P}_{TC}$ subsets in AIME24 ($|\mathbb{P}_{TC}| = 5$, $|\mathbb{P}_T| = 25$) and MATH-500 ($|\mathbb{P}_{TC}| = 42$, $|\mathbb{P}_T| = 458$), respectively. In both benchmarks, accuracy consistently drops for problems with procedural code.

## 2 RELATED WORKS

**Visual Geometric Perception** To study visual geometric reasoning, several benchmarks such as Euclid (Zhang et al., 2024a), MM-Math (Sun et al., 2024), GeoSense (Xu et al., 2025), MathVerse (Zhang et al., 2024b), and MathVista (Lu et al., 2023) have been introduced, each incorporating visual geometric content. These datasets measure large multi-modal models' comprehension of visual geometric concepts and their handling of mathematical problems with visual components. Their focus is mainly on diagram interpretation rather than procedural geometric code understanding, which represents a different but equally important aspect of geometric spatial reasoning.

**Mathematical Reasoning Benchmarks** A diverse array of benchmarks has been developed to evaluate the mathematical reasoning abilities of large language models (LLMs). Datasets such as

GSM8K (Cobbe et al., 2021), MATH-500 (Lightman et al., 2023), OlympiadBench (He et al., 2024), Minerva-MATH (Lewkowycz et al., 2022), CollegeMath (Tang et al., 2024), MMLU-STEM (Hendrycks et al., 2020), and AIME24 (MAA, 2025) primarily focus on algebraic, arithmetic, and word-problem reasoning. Many of these benchmarks target complex multi-step solutions, ranging from advanced high school mathematics to the level of international mathematical olympiads.

**Symbolic Graphics Perception and Generation**   A growing line of work targets the symbolic understanding and manipulation of SVG code for LLMs. Benchmarks such as SGP-Bench (Qiu et al., 2024), SVG Taxonomy (Xu & Wall, 2024), and SVGenius (Chen et al., 2025a) focus on SVG code parsing and perception, probing models' abilities to interpret, understand, and reason about symbolic vector graphics. Other efforts, such as SGP-GenBench (Chen et al., 2025b), provide benchmarks for evaluating models' capabilities in SVG code generation, while methods like Chat2SVG (Wu et al., 2025) propose approaches for improving SVG generation from natural language instructions. Together, these works lay important foundations for assessing both the comprehension and generative aspects of symbolic graphics in LLMs. However, there is still a lack of benchmarks specifically designed to test the construction of geometric spatial structures and deeper reasoning within geometry, which is the focus of GeoGramBench.

## 3   PROGRAM-TO-GEOMETRY

### 3.1   TASK DEFINITION

We define `Program-to-Geometry` as the task in which a model interprets procedural code to construct mathematical geometric representations, and subsequently reasons over these representations to solve geometry problems. This paradigm provides a comprehensive assessment of two fundamental capabilities: (a) the ability to accurately construct mathematical geometric diagrams from symbolic instructions, and (b) the ability to perform spatial reasoning and mathematical problem solving based on these constructed diagrams.

### 3.2   TAXONOMY

An effective taxonomy is critical for evaluating benchmark quality and pinpointing capability bottlenecks in the `Program-to-Geometry` task, which requires translating procedural geometric code into diagrammatic understanding. Traditional taxonomies, such as those based on topological complexity (Zhou et al., 2025), logical intricacy (Lin et al., 2025), or reasoning difficulty (e.g., high school to olympiad levels) (MAA, 2025; Sun et al., 2025; Hendrycks et al., 2021), focus on reasoning steps rather than the geometric structures central to this task. To address this, we propose a new taxonomy tailored to the `Program-to-Geometry` task, defined by the geometric complexity of diagrams derived from procedural code. This three-level hierarchy reflects the types and number of geometric elements:

- *Primitive Recognition (Primitive):* Problems involving procedural code that specify only one or two geometric primitives (e.g., points, lines, arcs, circles, polygons), focusing on basic mathematical properties such as length, area, or angle.

- *Local Relation Composition (Compositional):* Problems with multiple local geometric elements, requiring the recognition, integration, and composition of spatial relationships among subcomponents of the diagram.

- *Global Abstract Integration (Abstract):* Items demanding spatial direction, parameterization, recursion, 3D objects, composite structures, or advanced geometric operations (e.g., rotation, folding, projection), thus requiring not only the construction of complex diagrams but also global and stepwise spatial reasoning across the entire configuration.

To validate this taxonomy, we analyzed the QwQ-32B (Team, 2025a) performance on the MATH-500 (Lightman et al., 2023) dataset, comparing accuracy across reasoning complexity (per MATH-500 annotations) and geometry complexity for text-only ($\mathbb{P}_T$) and text+code ($\mathbb{P}_{TC}$) problems (see Figure 2). For $\mathbb{P}_T$, accuracy decreases with increasing reasoning complexity, consistent with existing benchmarks. In contrast, for $\mathbb{P}_{TC}$, accuracy is largely independent of reasoning complexity, with

significant drops tied to geometric intricacy. This trend, validated by a clear accuracy decline on MATH-500 as geometric complexity increases, confirms that geometric complexity, rather than reasoning steps, is the primary challenge in this task. Thus, our taxonomy provides a robust framework for evaluating model capabilities in `Program-to-Geometry` task.

### 3.3 TASK DATA FORMAT

Each problem consists of two parts: a textual description and geometric drawing code, such as written in Asymptote or Matplotlib style. The model receives both the text and the code as input. The target is a numeric answer, which may represent a length, area, volume, angle, ratio, or count. This setup requires the model to parse the drawing code, form an internal geometric representation, and reason through the mathematical question to produce the correct numeric solution.

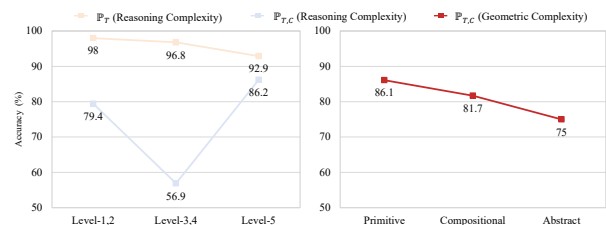

Figure 2: Effect of reasoning and geometric complexity.

### 3.4 RESEARCH QUESTIONS

Understanding the capabilities of large language models (LLMs) in the `Program-to-Geometry` task requires a systematic investigation into their ability to process and reason over procedural geometric code. This task demands a sequence of essential, hierarchically related skills: from recognizing basic geometric elements to composing complex spatial configurations and leveraging chain-of-thought (CoT) reasoning to enhance problem-solving. Investigating these capabilities is motivated by the need to identify specific bottlenecks in LLMs performance, guiding the development of models better equipped for spatial reasoning applications. Based on this task definition and taxonomy, we articulate the following research questions to structure our analysis of LLMs' behavior in the `Program-to-Geometry` task:

**RQ1:** *Is there evidence that LLMs can understand and represent basic geometric elements from program code?*

**RQ2:** *How effectively can LLMs compose and abstract geometric elements into coherent spatial configurations as specified by program code?*

**RQ3:** *How does CoT reasoning influence LLMs' spatial geometric reasoning abilities with program code?*

These research questions are fundamental to evaluating the progression of LLMs capabilities in this domain, and their investigation is critical for advancing our understanding of how to enhance symbolic-to-spatial reasoning in future models.

## 4 BENCHMARK CONSTRUCTION

In this section, we present the systematic construction process of **GeoGramBench**, a dedicated benchmark for `Program-to-Geometry` reasoning. We first introduce a critical challenge inherent to this task domain, namely answer leakage, before detailing our comprehensive data construction pipeline that forms the foundation of our benchmark (more details in Appendix C).

### 4.1 ANSWER LEAKAGE CHALLENGES

In the `Program-to-Geometry` task, a significant challenge arises from the potential for answer leakage within the code itself. The program code that generates geometric figures often contains precise numerical specifications that directly or indirectly reveal the answers sought. Benchmark like Math-500 (Lightman et al., 2023), we discovered numerous instances where answers were

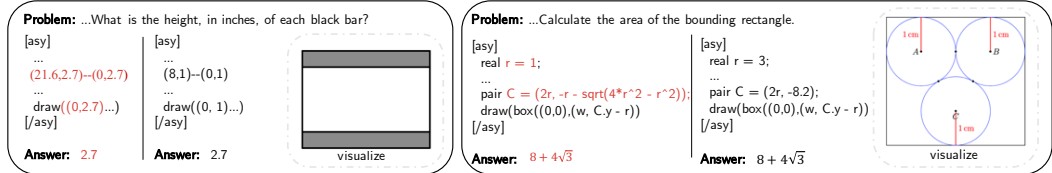

Figure 3: Illustration of two types of answer leakage in procedural code, highlighted in red. On the left is **direct leakage**, where the answer is explicitly given by a coordinate value in the Asymptote code; in this case, we rescale the coordinates to preserve the geometric shape. On the right is **indirect leakage**, where the answer can be computed from code parameters; here, we modify the procedural code to mask such critical information.

directly embedded in the Asymptote code. Similar issues persist across various open-source geometry problem collections we collected. As illustrated in Figure 3, we categorize two types of answer leakage in the procedural code. **Direct leakage** occurs when the answer is explicitly encoded as a coordinate value in the Asymptote code (e.g., a circle's radius or segment's length). **Indirect leakage** occurs when the answer can be computed from code parameters or formulas.

## 4.2 COLLECTION AND PREPROCESSING

We first aggregated approximately 905K candidate problems from three open-source mathematics datasets, including NuminaMath-1.5 (Li et al., 2024), HARP (Yue et al., 2024), and Omni-MATH (Gao et al., 2024), with a focus on sources rich in geometry content. We filtered for problems containing embedded Asymptote code by searching for [asy] and [/asy] tags, resulting in a subset comprising about 1% (9,260 problems). We then deduplicated this subset using an $n$-gram ($n = 8$) similarity approach (Muennighoff et al., 2025), reducing the set to 1,782 unique items. Finally, by following the schema from s1 (Muennighoff et al., 2025) and leveraging GPT-4o (Hurst et al., 2024) for prompt-based classification, we selected only geometry problems, yielding 1,247 geometry-focused items for subsequent curation.

## 4.3 HUMAN REFINEMENT AND VERIFICATION

To ensure data quality and suitability for geometry code understanding tasks, we implemented a two-stage manual verification process, conducted by a team of four experts (each holding a master's degree or higher in mathematics or related fields). The first round aimed to standardize problem types and formats, while the second round focused on enhancing overall problem quality.

In the **first round**, we performed initial screening and format normalization: (a) non-relevant questions (such as hyperlink chains, multi-part items, and proofs) were filtered out according to best practices from BigMath (Albalak et al., 2025); (b) convertible multiple-choice questions were transformed into open-form computation problems by removing options, while those not amenable to conversion were discarded entirely; and (c) answers were standardized into consistent LATEX format. At the end of this screening, 547 candidate problems remained.

In the **second round**, we implemented a rigorous three-pronged refinement process to improve problem quality:

- Decontamination: To minimize community-sourced contamination, we systematically revised problem statements by removing redundant descriptive information that might enable direct textual inference. Additionally, we adjusted problem conditions and modified corresponding answers to maintain mathematical consistency. Furthermore, we adjusted the answer requirements (such as replacing queries about lengths with those about area, volume, or ratios) to further reduce the risk of leakage and promote authentic geometric reasoning.

- Answer Leakage Prevention: As detailed in Section 4.1, to address this task-specific vulnerability, we implemented two targeted strategies: systematically rescaling coordinates while preserving geometric relationships for direct leakage, and modifying or masking code parameters for indirect leakage. These interventions ensure that answers cannot be derived through mere code inspection (see Figure 3).

- Accuracy Verification: Each answer was manually checked for correctness; items with ambiguous, unverifiable, or doubtful solutions were removed.

Through this thorough process, we ultimately obtained 392 high-quality, contamination-free geometry problems for augmentation and evaluation.

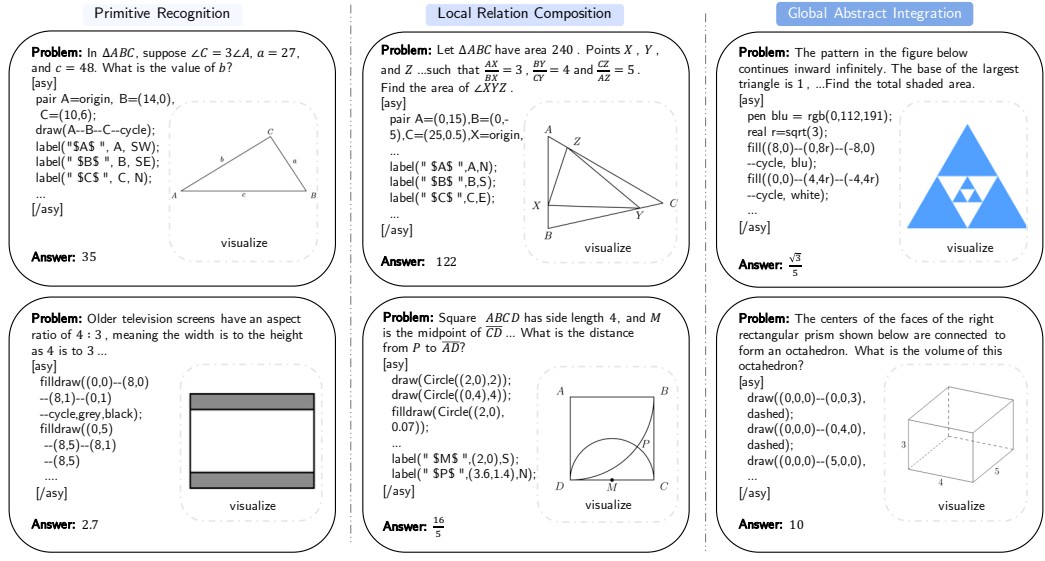

Figure 4: Representative examples from GeoGramBench illustrating the three ascending `Program-to-Geometry` difficulty levels: *Primitive Recognition*, *Local Relation Composition*, and *Global Abstract Integration*. Each category is exemplified by two sampled problems, highlighting the increasing spatial complexity and abstraction across levels.

## 4.4 BENCHMARK AUGMENTATION

To enhance difficulty balance and problem diversity, we supplemented GeoGramBench with additional items: 5 geometry problems from AIME24 (MAA, 2025), 42 from MATH-500 (Lightman et al., 2023), and 61 geometric problems adapted from Mathverse (Zhang et al., 2024b). For the Mathverse subset, we selected representative solid geometry problems and manually transcribed diagrams into `matplotlib` code to diversify the procedural drawing code within the dataset. Our experiments indicate minimal impact from the choice of drawing language (see Appendix A). Altogether, GeoGramBench comprises 500 geometry problems, supporting robust evaluation across a variety of geometric phenomena.

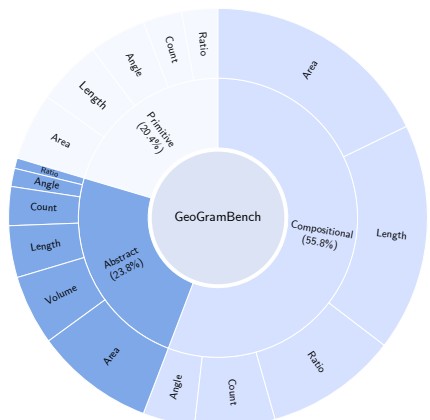

Figure 5: Statistics of GeoGramBench.

## 4.5 DIFFICULTY AND SUBTYPE CATEGORIZATION

Building on our theoretical and empirical insights in Section 3.2, we categorize all 500 GeoGram-Bench problems into three ascending difficulty levels: *Primitive Recognition*, *Local Relation Composition*, and *Global Abstract Integration*, based on the type and number of geometric elements and the spatial relationships involved (see Figure 4). The categorization is implemented through a combination of GPT-4o (Hurst et al., 2024) assisted classification and thorough human expert review.

The resulting distribution, detailed in Figure 5, establishes GeoGramBench as the largest and most diverse benchmark for the `Program-to-Geometry` task to date.

To facilitate a deeper analysis of LLMs performance and failure modes, we further categorize these problems based on problem-solving objectives, identifying common challenges across geometric properties. This subtype classification divides the three levels into six task types: angle, length, area, volume, ratio, and count, determined via manual annotation. Figure 5 illustrates the distribution of GeoGramBench's 500 problems, highlighting the diverse representation of multiple task types within each category, including the introduction of volume tasks in the most complex level, enabling targeted investigation into specific reasoning difficulties (e.g., 3D structures or angle computations) to guide future model improvements. Detailed subtype statistics and definitions are provided in Appendix C.8, underscoring GeoGramBench's value as a diagnostic tool for evaluating LLMs capabilities across this varied range of problem types.

## 5 EXPERIMENT

We benchmark 19 popular LLMs on GeoGramBench, providing a broad comparative analysis in this section. Section 5.1 details our evaluation framework and prompt engineering strategies. Section 5.2 introduces the tested models, followed by quantitative comparisons in Section 5.3.

### 5.1 EVALUATION PROTOCOLS

For open-source models, we adopt the Luo et al. (2025) framework for evaluation, while for closed-source models, we utilize official APIs with identical prompt templates (*Let's think step by step and output the final answer within \boxed{}.*). All result parsing is standardized using Luo et al. (2025), with assistance from GPT-4o when necessary. Each problem is evaluated in a zero-shot setting: the model input consists strictly of the problem text and the procedural geometry drawing code. For each problem instance, we sample 8 responses using temperature 0.6, and report final accuracy as the mean over these 8 outputs, which balances model stochasticity and answer reliability.

### 5.2 EVALUATION MODELS

We evaluate a total of 19 mainstream LLMs, including both proprietary APIs and leading open-source systems. The closed-source models include GPT-5 (OpenAI, 2025), GPT-4o (Hurst et al., 2024), GPT-o3-mini (Pfister & Jud, 2025), the GPT-o1 series (Jaech et al., 2024), and Gemini-Pro-1.5 (Team et al., 2023). The open-source models cover a wide range of scales, including DeepSeek-R1 (Guo et al., 2025), DeepSeek-v3-0324 (Liu et al., 2024), Qwen3 (Team, 2025b) and QwQ-32B (Team, 2025a), as well as other prominent models from 32B down to 1.5B parameters: DeepSeek-R1-Distill variants (Guo et al., 2025), Bespoke-Stratos-32B (Labs, 2025), s1.1-32B (Muennighoff et al., 2025), Sky-T1-mini-7B (Li et al., 2025), and DeepScaleR-1.5B-preview (Luo et al., 2025).

### 5.3 MAIN RESULTS

**Closed-source models outperform open-source models.** Table 1 shows that the latest GPT-5 model achieves state-of-the-art performance, with an overall average accuracy of 75.01%. GPT-5 also obtains the highest scores in both the *Primitive* and *Compositional* levels. Among open-source models, Qwen3-235B-Thinking-2507 ranks second, surpassing earlier closed-source models like GPT-o1 and GPT-o3-mini, with an average accuracy of 74.00%. GPT-o1 and GPT-o3-mini follow, each with average accuracies around 70%.

**All models perform below 50% in the Abstract level.** As the geometric program complexity increases, accuracy drops sharply for all models on the *Abstract* level, with no model surpassing 50%. For example, Qwen3-235B-Thinking-2507 achieves 89.09% accuracy on the *Primitive* level and 79.12% on the *Compositional* level, but only 49.05% on the *Abstract* level. Similarly, GPT-5 reaches 90.44% on the *Primitive* level and 84.59% on the *Compositional* level, but falls to just 39.26% on the *Abstract* level. This trend underscores significant limitations in current LLMs' ability to construct and reason about complex geometric figures and spatial relationships, particularly when deeper geometric abstraction is required.

**Most challenging subtypes: angle and volume.** At the *Primitive* and *Compositional* levels, which focus on 2D geometric figures, the angle subtype is most challenging, as it requires the model to reconstruct and reason about implicit spatial relationships in 2D, which is difficult with only code input. In contrast, length or count subtypes are typically more straightforward, often solvable with direct algebraic or counting methods. At the *Abstract* level, dominated by 3D geometric figures, the most difficult subtypes shift to area and especially volume. Area and volume problems require complete 3D spatial understanding, making them especially difficult at the *Abstract* level. This means models struggle much more with these tasks. Angle problems can sometimes be simplified to 2D scenarios using vector or triangle-based techniques, so their relative difficulty is lower compared to area and volume. The key point is not that angle questions become simple, but that area and volume tasks in 3D geometry present a much greater challenge for current models. Thus, the stark accuracy drop for area and volume in the *Abstract* level reflects the greater demand for true 3D spatial integration and highlights the significant challenges that current models face in higher-dimensional reasoning.

| Model | Primitive | | | | | | Compositional | | | | | | Abstract | | | | | | | ALL |
|---|---|---|---|---|---|---|---|---|---|---|---|---|---|---|---|---|---|---|---|---|
| | Avg. | Angle | Length | Area | Ratio | Count | Avg. | Angle | Length | Area | Ratio | Count | Avg. | Angle | Length | Area | Volume | Ratio | Count | |
| *Closed-source Models* | | | | | | | | | | | | | | | | | | | | |
| GPT-5 | **90.44** | **85.80** | 87.50 | **93.75** | 87.50 | **99.17** | **84.59** | **74.02** | **86.65** | **88.20** | 77.50 | **90.32** | 39.26 | **70.91** | 80.67 | 19.29 | 21.76 | 53.13 | **80.67** | **75.01** |
| GPT-o3-mini | 83.49 | 69.57 | 87.50 | 92.24 | 76.47 | 89.47 | 76.10 | 51.96 | 78.91 | 77.37 | 75.42 | 79.55 | 42.67 | 55.36 | 59.56 | 30.14 | 30.60 | 59.38 | 71.32 | 70.00 |
| GPT-o1 | 85.92 | 71.15 | **88.39** | 92.67 | 88.23 | 90.13 | 76.12 | 50.00 | 80.63 | 79.60 | 69.07 | 79.55 | 44.67 | 64.28 | 58.15 | 35.30 | **36.20** | **62.50** | 57.78 | 70.92 |
| GPT-o1-preview | 73.95 | 57.21 | 84.82 | 80.17 | 70.59 | 74.34 | 55.87 | 36.41 | 56.68 | 57.08 | 58.69 | 57.58 | 25.33 | 48.21 | 29.89 | 15.17 | 23.71 | 53.13 | 37.5 | 53.15 |
| GPT-o1-mini | 78.89 | 68.30 | 86.16 | 84.91 | 74.26 | 77.63 | 63.31 | 43.48 | 65.89 | 68.68 | 59.66 | 57.20 | 27.14 | 34.14 | 43.48 | 15.44 | 20.69 | 31.25 | 49.26 | 58.94 |
| GPT-4o | 40.02 | 25.48 | 46.43 | 47.84 | 38.23 | 40.13 | 21.36 | 9.78 | 23.00 | 21.93 | 22.03 | 20.45 | 4.51 | 14.29 | 6.52 | 0.70 | 2.10 | 25.00 | 8.82 | 21.40 |
| Gemini-Pro-1.5 | 48.77 | 54.81 | 49.50 | 54.81 | 47.32 | 53.33 | 31.41 | 20.00 | 29.97 | 32.30 | 34.07 | 35.89 | 14.39 | 19.64 | 23.13 | 9.78 | 4.17 | 28.13 | 29.17 | 31.64 |
| *Open-source Models* | | | | | | | | | | | | | | | | | | | | |
| Qwen3-235B-Thinking-2507 | 89.09 | 84.10 | 88.00 | 90.38 | **91.07** | 94.17 | 79.12 | 68.75 | 81.68 | 83.71 | 68.38 | 83.06 | **49.05** | 60.71 | 63.75 | **44.84** | 31.48 | 50.00 | 68.33 | 74.00 |
| DeepSeek-R1 | 84.68 | 73.86 | 87.50 | 87.98 | 89.29 | 85.83 | 75.13 | 66.80 | 76.85 | 78.93 | 70.10 | 72.98 | 40.86 | 60.71 | 53.75 | 34.78 | 24.07 | 59.38 | 58.33 | 69.17 |
| DeepSeek-v3-0324 | 79.73 | 66.35 | 87.50 | 81.90 | 80.89 | 82.23 | 68.71 | 52.72 | 71.85 | 73.58 | 64.19 | 60.98 | 28.29 | 51.79 | 45.65 | 16.51 | 18.53 | 56.25 | 41.91 | 62.05 |
| Qwen3-32B | 86.89 | 82.39 | 87.50 | 84.13 | 91.96 | 92.50 | 75.18 | 60.00 | 78.12 | 81.32 | 63.48 | 78.23 | 43.28 | 66.07 | 50.62 | 37.50 | 24.54 | 68.75 | 67.5 | 69.98 |
| QwQ-32B | 85.17 | 75.57 | 86.50 | 82.21 | 90.18 | 90.83 | 73.12 | 51.88 | 77.70 | 79.63 | 65.93 | 69.35 | 37.92 | 44.64 | 53.23 | 31.25 | 25.46 | 46.88 | 59.17 | 67.12 |
| DeepSeek-Distill-Qwen-32B | 79.78 | 67.61 | 83.50 | 77.88 | 83.93 | 90.00 | 67.83 | 51.88 | 69.03 | 75.28 | 59.07 | 67.34 | 35.92 | 50.00 | 48.13 | 26.90 | 21.76 | 53.13 | 60.00 | 62.68 |
| Bespoke-Stratos-32B | 62.50 | 40.34 | 72.50 | 62.50 | 71.43 | 70.00 | 42.56 | 33.13 | 42.05 | 46.35 | 43.38 | 37.90 | 17.02 | 28.57 | 26.25 | 8.97 | 15.28 | 46.88 | 19.17 | 40.55 |
| s1.1-32B | 75.37 | 50.57 | 84.00 | 77.40 | 80.00 | 80.00 | 58.96 | 39.38 | 61.65 | 62.50 | 57.11 | 56.88 | 26.58 | 33.93 | 40.00 | 16.58 | 18.52 | 43.75 | 45.83 | 54.60 |
| DeepSeek-Distill-Qwen-7B | 72.79 | 60.80 | 82.50 | 67.79 | 75.00 | 80.83 | 58.74 | 40.62 | 58.10 | 67.56 | 53.68 | 55.24 | 24.16 | 33.93 | 40.62 | 11.68 | 18.52 | 31.25 | 44.17 | 53.38 |
| Sky-T1-mini-7B | 71.45 | 57.95 | 79.50 | 64.42 | 75.89 | 85.83 | 57.75 | 40.00 | 58.81 | 64.61 | 51.23 | 57.26 | 24.79 | 35.71 | 38.75 | 13.32 | 18.06 | 34.38 | 45.83 | 52.70 |
| DeepSeek-Distill-Qwen-1.5B | 60.29 | 48.86 | 76.50 | 55.29 | 62.5 | 56.67 | 39.02 | 21.25 | 41.19 | 46.49 | 32.11 | 34.27 | 11.03 | 7.14 | 19.38 | 4.62 | 9.26 | 21.88 | 21.67 | 36.70 |
| DeepScaleR-1.5B-preview | 65.44 | 52.27 | 80.50 | 57.21 | 69.64 | 70.00 | 47.89 | 30.63 | 51.14 | 53.09 | 42.16 | 44.35 | 15.76 | 12.50 | 16.25 | 9.78 | 16.67 | 21.88 | 31.67 | 43.83 |

Table 1: Accuracy (%) of selected closed-source and open-source LLMs on GeoGramBench across three difficulty levels. For each model, the lowest-performing subtype within each level is highlighted with a background color (⬛), and for each subtype, the highest accuracy among all models is shown in **bold**.

## 6 Behavior Analysis of LLMs

We address our RQs through both quantitative and qualitative analyses based on benchmarking results and model responses. Furthermore, we summarize several common failure patterns observed across different models.

**RQ1:** *Is there evidence that LLMs can understand and represent basic geometric elements from program code?*

RQ1 investigates the fundamental ability of LLMs to recognize basic geometry elements, which can be quantitatively measured by the evaluation results of Primitive Recognition.

As shown in Table 1, most of the models achieve 60% accuracy on the *Primitive Recognition* level, suggesting that they can effectively parse and build basic geometric scenes from procedural codes. Qualitatively, some of the model responses explicitly reveal the capability to interpret and reconstruct geometric information. As shown in Figure 6, models frequently examine the procedural code for geometry understanding: "Now, looking at the Asymptote code", "Let me parse the Asymptote code a bit", and "maybe I should try to visualize this". They can also identify simple geometric relationships according to the procedural code. For example, "c is (2,0), so c/2 is (1,0). So the inner arc is between points a/2 and c/2", and "path inner = arc(d, a/2, c/2, CW);...path outer = arc(d, c, a, CCW);". These behavior demonstrate that LLMs are intent and capable to map procedural code into internal geometric structures. In conclusion, modern LLMs are able to construct basic geometric representations from procedural code.

**RQ2:** *How effectively can LLMs compose and abstract geometric elements into coherent spatial configurations as specified by program code?*

RQ2 investigates LLMs' capability of the geometry composition and global representation abstraction. According to the results in Table 1, all models experience a significant drop in accuracy from *Compositional* problems to *Global Abstract Integration*. For example, GPT-o1 drops from 76.12% to 44.67%, and DeepSeek-R1 drops from 75.13% to 40.86%. These results indicate that current LLMs may lack of compositional and spatial abstraction ability to solve complex geometry problems. Qualitatively, while models can often parse and assemble some local structures, small errors in local constructions frequently appear, preventing LLMs to construct a complete and coherent global representation. As illustrated in Figure 6, a model may read a piece of code like "path inner = arc(d, a/2, c/2, CW)" and reason about directions ("which would be the other direction compared to the inner counterclockwise path before"), but a single mistake in local spatial assignment may generate downstream confusion: "maybe I got the direction of the angle wrong?... the actual angle between the points is \theta, so the area calculations still hold.". This phenomena suggests that modern LLMs may not good at capturing complex compositional geometry relationships for high level spatial reasoning. In summary, although LLMs have made progress in local geometric parsing, their ability to synthesize and reason over globally consistent spatial structures in `Program-to-Geometry` tasks remains limited.

**RQ3:** *How does CoT reasoning influence LLMs' spatial geometric reasoning abilities with program code?*

Qualitatively, while models frequently perform iterative self-reflection and verification of code ("Let me check again"), and repeatedly parse diagram instructions, their CoT trajectories rarely correct or update internal geometric understanding, as shown in Figure 6. For instance, the model may cycle through algebraic steps and verbalize uncertainty ("Hmm, this is a bit confusing without seeing the actual diagram. Since I can't see the diagram, maybe I should proceed with the information given."), yet consistently fails to resolve spatial relationships or integrate local shapes into a whole. This observation illustrates that CoT may lead LLMs fall into repetitive symbolic reasoning. Such repetitiveness does not beneficial for LLMs to construct high level spatial representations as a whole, even leading to confusion about complex geometry relationships.

Quantitatively, our experimental results show a downward trend in accuracy as geometric complexity increases, consistent with our benchmark taxonomy's emphasis on geometric intricacy rather than reasoning steps. This suggests that most LLMs exhibit weak internal geometric representation capabilities, and long CoT reasoning provides limited benefit for improving this aspect. To further investigate this, we conducted an quantitative experiment using Token Budget Forcing (BF) (Muennighoff et al., 2025), detailed in Appendix E, which extends CoT reasoning and provides a robust validation of its limitations. These findings highlight a critical bottleneck: while CoT enhances mathematical reasoning in other domains, its effectiveness in `Program-to-Geometry` tasks is constrained by LLMs' weak spatial abstraction capabilities.

**Common Failure Patterns**   Through extensive qualitative analysis of model responses, conducted by manually reviewing a substantial number of failure cases, we distilled several prevalent failure patterns that are shared across multiple LLMs. Due to the current lack of accurate automated assessment methods for this process, our analysis is based on representative examples rather than exhaustive annotation. Nevertheless, these patterns foreground critical limitations in current model behavior. (1) The models exhibit a pronounced preference for algebraic methods, frequently relying on coordinate calculations rather than leveraging geometric constructions, even when the latter would be significantly more efficient. This algebraic bias often results in the neglect of essential geometric heuristics. (2) LLMs rarely introduce auxiliary lines or points, a classic strategy in geometry that can simplify complex problems, indicating limited flexibility in exploring alternative geometric solutions. (3) Models consistently struggle with instructions that involve spatial orientation, such as distinguishing between clockwise and counterclockwise directions or interpreting vertical and horizontal references. These difficulties often lead to misinterpretation or incorrect reasoning steps. (4) A common issue is the confusion in mapping symbolic relationships (e.g., angle or vertex labels) to their corresponding geometric elements in the diagram, reflecting an insufficient internal spatial representation and a lack of robust grounding for abstract symbols.

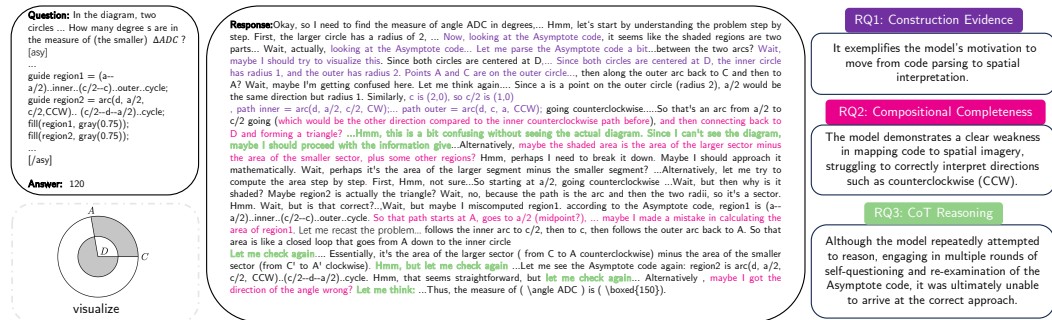

Figure 6: Illustrative solution process generated by the QwQ-32B model on a *Local Relation Composition* problem. The model initially attempts to construct spatial representations from the provided code, then interprets geometric elements such as direction and region, exhibiting behavior aligned with all three research questions (RQ1–RQ3): local construction, compositional integration, and chain-of-thought-based refinement. Multiple rounds of reflection and verification are observed, although these iterative steps do not consistently yield correct or fully integrated solutions.

Overall, our behavior analysis offers an in-depth diagnosis of how contemporary LLMs process programmatic geometric information and where their reasoning strategies fall short. By systematically linking model behaviors to distinct task attributes, we not only clarify the boundaries of current capabilities, but also map out the critical obstacles restricting progress in symbolic-to-spatial reasoning.

# 7 CONCLUSION

In this work, we present **GeoGramBench**, the first large-scale benchmark for evaluating LLMs on the challenging `Program-to-Geometry` task, which connects procedural code with geometric reasoning. Our experiments on 19 leading models show that even the strongest models achieve less than 50% accuracy on complex problems, revealing critical gaps in symbolic to spatial understanding. Through a new taxonomy and comprehensive analysis, we identify major shortcomings and establish GeoGramBench as an important resource for advancing this research direction. We hope this benchmark will encourage further progress toward models with better spatial abstraction and symbolic reasoning, pushing forward the development of AI with stronger spatial intelligence.

## ETHICS STATEMENT

This work introduces a mathematical reasoning benchmark and does not involve human subjects, personal data, sensitive information, or real-world deployments. The dataset is constructed purely from publicly available mathematical problem sources and procedural code, with rigorous data cleaning and verification to ensure research integrity. We are not aware of any ethical risks or concerns related to privacy, fairness, safety, discrimination, or legal compliance in this study.

## REPRODUCIBILITY STATEMENT

The complete details of GeoGramBench dataset construction are provided in Section 4 and Appendix C. We include all evaluation code in the supplementary materials, along with detailed instructions for running experiments. For model evaluation, we access closed-source models via their official APIs, and open-source models are obtained from Hugging Face (https://huggingface.co/), complying with their respective terms of use.

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

## A   EFFECT OF DRAWING LANGUAGE ON PROGRAM-TO-GEOMETRY PERFORMANCE

A key motivation for our investigation is to determine to what extent challenges in `Program-to-Geometry` reasoning arise from the logic of geometric construction itself, rather than from surface-level code syntax or unfamiliarity with specific drawing languages. To test this, we translated 5 geometry questions containing Asymptote code from AIME24 and 42 questions from MATH-500 into equivalent Python `matplotlib` code, holding geometric content constant while varying only the programmatic language. As shown in Figure 7, QwQ-32B exhibits less than 1% difference in absolute accuracy between the Asymptote and Matplotlib versions on both benchmarks. This minimal gap provides strong evidence that the principal bottleneck in `Program-to-Geometry` task performance is not due to the choice of drawing language, but rather stems from deeper difficulties in spatial abstraction and geometric reasoning from code. This result reinforces our conclusion that surface syntax is not the main limiting factor for LLMs in this domain.

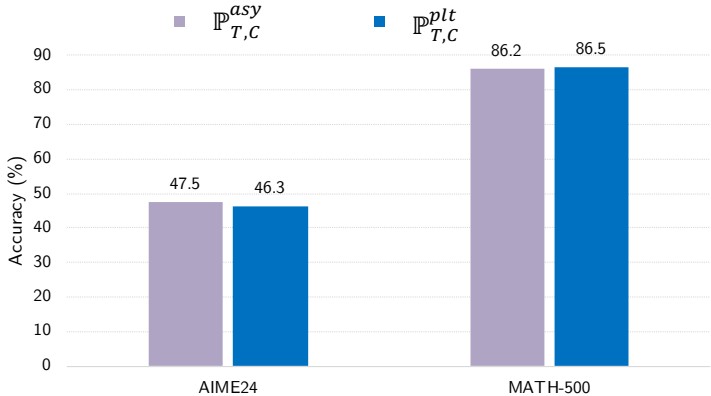

Figure 7: Comparison of QwQ-32B accuracy on equivalent geometry problems expressed in Asymptote versus Matplotlib code (AIME24 and MATH-500). The negligible performance gap demonstrates that `Program-to-Geometry` capability is independent of drawing language syntax.

**Example**

**Problem Statement:**
Rectangles $ABCD$ and $EFGH$ are drawn such that $D, E, C, F$ are collinear. Also, $A, D, H, G$ all lie on a circle. If $BC = 16$, $AB = 107$, $FG = 17$, and $EF = 184$, what is the length of $CE$?

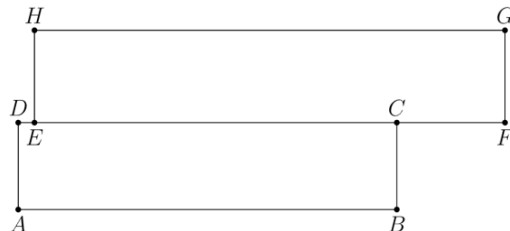

Figure 8: Visualization generated from the drawing code

**Drawing Code (Asymptote):**

```
import graph;
unitsize(0.1cm);
pair A = (0,0);
pair B = (70,0);
pair C = (70,16);
pair D = (0,16);
```

```
pair E = (3,16);
pair F = (90,16);
pair G = (90,33);
pair H = (3,33);
dot(A^^B^^C^^D^^E^^F^^G^^H);
label("\$A\$", A, S);
label("\$B\$", B, S);
label("\$C\$", C, N);
label("\$D\$", D, N);
label("\$E\$", E, S);
label("\$F\$", F, S);
label("\$G\$", G, N);
label("\$H\$", H, N);
draw(E--D--A--B--C--E--H--G--F--C);
```

**Drawing Code (Matplotlib):**

```
import matplotlib.pyplot as plt

A = (0, 0)
B = (70, 0)
C = (70, 16)
D = (0, 16)
E = (3, 16)
F = (90, 16)
G = (90, 33)
H = (3, 33)

for pt in [A, B, C, D, E, F, G, H]:
plt.plot(pt[0], pt[1], 'ko')

plt.text(A[0], A[1]-1, "\$A\$", ha='center', va='top', fontsize
    =13)
plt.text(B[0], B[1]-1, "\$B\$", ha='center', va='top', fontsize
    =13)
plt.text(C[0], C[1]+1, "\$C\$", ha='center', va='bottom',
    fontsize=13)
plt.text(D[0], D[1]+1, "\$D\$", ha='center', va='bottom',
    fontsize=13)
plt.text(E[0], E[1]-1, "\$E\$", ha='center', va='top', fontsize
    =13)
plt.text(F[0], F[1]-1, "\$F\$", ha='center', va='top', fontsize
    =13)
plt.text(G[0], G[1]+1, "\$G\$", ha='center', va='bottom',
    fontsize=13)
plt.text(H[0], H[1]+1, "\$H\$", ha='center', va='bottom',
    fontsize=13)

plt.plot([E[0], D[0], A[0], B[0], C[0], E[0]], [E[1], D[1], A
    [1], B[1],
C[1], E[1]], color='black')
plt.plot([E[0], H[0], G[0], F[0], C[0]], [E[1], H[1], G[1], F
    [1], C[1]], color='black')

plt.xlim(-5, 95)
plt.ylim(-5, 38)
plt.gca().set_aspect('equal')

plt.axis('off')
```

```
    plt.tight_layout()
    plt.show()
```

## B  PREVENTING INFORMATION LEAKAGE IN PROCEDURAL GEOMETRY CODE

A critical aspect of dataset curation for `Program-to-Geometry` evaluation is the prevention of information leakage through the procedural drawing code. In this context, information leakage refers to situations where the answer to a geometry problem is either explicitly or implicitly encoded in the program, enabling a model (or human) to bypass genuine geometric reasoning and instead extract the solution directly from code inspection.

We identify two primary forms of leakage:

- **Direct leakage**: The answer appears explicitly in the code, for example as a coordinate, length, or parameter value (e.g., a circle radius or segment described directly in the Asymptote code).

- **Indirect leakage**: The answer can be inferred by performing simple calculations or extracting formula results from the parameters or structure of the code, even though it is not written verbatim.

To mitigate these risks, we systematically reviewed all procedural code in the dataset. For direct leakage, critical coordinates and parameters are rescaled or randomized while preserving the diagram's structure. For indirect leakage, problem variables and code formulas are modified or masked to preclude simple reverse engineering of the answer.

Below we present concrete examples comparing original and mitigated code for selected problems. Each example includes its problem statement and paired Asymptote code, annotated as "before" and "after" modification.

**Example 1:**

**Problem Statement:**
In $\triangle ABC$, point $F$ divides side $AC$ in the ratio $1 : 2$. Let $E$ be the point of intersection of side $BC$ and $AG$ where $G$ is the midpoint of $BF$. The length of $EC$ divided by the length of $BE$ is ?

**Answer: 3**

**Before modification (Leakage present):**

```
size(2.5inch);
pair A, B, C, E, F, G;
A = (0,3);
B = (-1,0);
C = (3,0);
E = (0,0);
F = (1,2);
G = intersectionpoint(B--F,A--E
    );
draw(A--B--C--cycle);
draw(A--E); draw(B--F);
label(\"$A$\",A,N);
label(\"$B$\",B,W);
label(\"$C$\",C,dir(0));
label(\"$E$\",E,S);
label(\"$F$\",F,NE);
label(\"$G$\",G,SE);
```

**After modification (Leakage mitigated):**

```
size(2.5inch);
pair A, B, C, E, F, G;
A = (0,3);
B = (-1,0);
C = (4,0);
E = (0,0);
F = (1.14, 2.14);
G = intersectionpoint(B--F,A--E);
draw(A--B--C--cycle);
draw(A--E); draw(B--F);
label(\"$A$\",A,N);
label(\"$B$\",B,W);
label(\"$C$\",C,dir(0));
label(\"$E$\",E,S);
label(\"$F$\",F,NE);
label(\"$G$\",G,SE);
```

Figure 9: Side-by-side comparison of Asymptote code: before (left) and after (right) information leakage mitigation.

**Example 2:**

**Problem Statement:**
In rectangle $ABCD$, point $M$ is the midpoint of $\overline{AD}$. The area of $\triangle AMC$ is 12, and $\frac{AD}{AB} = \frac{3}{2}$. Find the length of side $AD$.

**Answer: 8**

**Before modification (Leakage present):**

```
size(4cm);
draw((0,4)-(0,0)-(6,0)-(6,8)
-(0,8)-(0,4)-(6,8)-(0,0));
label(\"$A$\", (0,0), SW);
label(\"$B$\", (6, 0), SE);
label(\"$C$\", (6,8), NE);
label(\"$D$\", (0, 8), NW);
label(\"$M$\", (0, 4), W);
```

**After modification (Leakage mitigated):**

```
size(4cm);
draw((0,2)--(0,0)--(3,0)--(3,4)
--(0,4)--(0,2)--(3,4)--(0,0));
label("$A$", (0,0), SW);
label(\"$B$\", (3, 0), SE);
label(\"$C$\", (3,4), NE);
label(\"$D$\", (0, 4), NW);
label(\"$M$\", (0, 2), W);
```

Figure 10: Side-by-side comparison of Asymptote code: before (left) and after (right) information leakage mitigation.

## C    DETAILED BENCHMARK CURATION

We assemble a team of four experts (each holding a Master's degree or higher in mathematics or related fields) to ensure data quality. Our team manually verifies and refines samples from three aspects: question reformulation and standardization, decontamination, answer verification and leakage prevention.

### C.1    INTRODUCTION TO ASYMPTOTE CODE

Asymptote is a vector graphics language designed for technical drawing and geometric visualization. "Asymptote code" refers to structured source code written in the Asymptote language, which describes geometric primitives (such as points, lines, circles, polygons, 2D and 3D shapes), spatial relationships, and drawing instructions. The Asymptote syntax supports mathematical expressions, procedural constructs, and graphical commands, enabling precise programmatic construction of geometric diagrams.

### C.2    STANDARDIZATION OF PROCEDURAL CODE

Our standardization pipeline for procedural code involves the following key steps:

1. Initial Filtering: We collect a large set of mathematical problems and filter for those containing embedded Asymptote code by searching for [asy] and [/asy] tags. This step isolates problems that include geometry drawing code.

2. Syntax Validation: We use the Asymptote tool to check for syntax correctness in each drawing code snippet. Problems with invalid syntax or unrenderable diagrams are excluded.

3. Diagram Verification and Adjustment: We confirm that each code snippet produces a valid and clear geometric diagram in Asymptote. If necessary, we adjust canvas size and proportions to avoid visual overlap, ambiguity, or unclear geometric elements.

4. Answer Leakage Prevention: As described in Section 4.1, we systematically review each code segment for direct or indirect answer leakage. Direct leakage (where answers appear explicitly in the code) is handled by rescaling coordinates or revising geometry. Indirect leakage (where answers can be inferred from code parameters) is prevented by masking or modifying the relevant parts of the code to ensure the answer cannot be obtained by inspection alone.

### C.3 QUESTION REFORMULATION AND ANSWER STANDARDIZATION

**Question reformulation** The formulation of each sample in GeoGramBench should be simple QA pairs for convenient evaluation. To achieve this, we start to deal with multiple choice questions, proof-based questions and multi-part problems, which are not in QA format. Multiple choice questions can be transformed into open-ended computation problems by preserving the correct choice as the answer and removing all other choices. Some of the proof-based questions can be transformed into computation problems (like "Prove that $PA = 4PB$" can be rewrite to "Compute the ratio between $PA$ and $PB$"), whereas others are not suitable for such transformation (like "Prove that $AB \geq 3PR$). Multi-part problem always consists of several sub-problems, which can be simplified into a single question format by retaining one of the computable sub-questions. Questions amenable to conversion can be retained and reformulated into new QA samples, while others may be excluded from the benchmark. According to the aforementioned rules, our team members carefully assess the formulation of each question and perform corresponding modifications and deletion.

**Answer standardization** Considering the diversity and complexity of mathematical expressions, answer standardization is crucial for accurately evaluating model-generated responses. Our team manually modify the answer of each question by removing arithmetic operators (like $+$, $-$), letters and characters that irrelevant for computation and evaluation (like \text{cm^2}), and standardize each answer into LaTeX format as simple as possible (like simplify \frac{28}{\sqrt{7}}} to 4\sqrt{7}). The above operations successfully ensure the consistency of question formulation and answer standardization, which benefits subsequent data processing and contributes reliable benchmarking. The resulting subset contains 547 candidate samples.

### C.4 DECONTAMINATION

Most of the samples we collected originates from public datasets and internet resources, which indicates a high possibility that these data has already been included in the LLMs' pre-training corpora. Besides, current data samples contains a certain degree of redundancy and unnecessary information, which may introduce unexpected bias to benchmarking. To mitigate the above influences as much as possible, our team manually perform data decontamination for all the 547 samples from three aspects:

**Extraneous information removal** We believe hyperlinks and code comments are not only unnecessary information for mathematic geometry spatial reasoning, but also introduce text bias for mathematic geometry problem reasoning. As a result, each member in our team carefully examine and delete all these contents in each question;

**Problem statement rephrasing**

To prevent samples from being solved solely based on question statement, encourage LLMs focus on mathematic geometry spatial reasoning, we reduce some comprehensive and specific mathematical expressions in question text. To minimize the overlap between LLMs' pre-training corpora and benchmarking samples, our team modifies the given condition and question objective of some samples;

**Coordinate modification** In some samples, the coordinates used to generate pictures are identical to the given conditions in the problem statement, which may enable LLMs to derive answer through algebraic geometry reasoning based on text solely. Such problem solving approach cannot effectively evaluate the mathematic geometry spatial reasoning ability of LLMs. To decrease the possibility of LLMs using algebraic geometry problem solving approach, we adjust the coordinates in each samples program code, which maintains the geometric shape and relationship of the original picture. The above decontamination methods ensures each item in GeoGramBench is a completely new sample, contributing to valuable and reliable mathematic geometry spatial reasoning benchmarking.

### C.5 ANSWER VERIFICATION AND LEAKAGE PREVENTION

**Answer verification** We observe that some of the original answers are wrong to the corresponding questions after decontamination. To avoid such circumstances, we carefully verify the answer of each sample one by one by both referencing the original question from the Internet and calculate answer by ourselves. The QA pairs that cannot be searched on the Internet are removed.

**Answer leakage prevention** We find some of the correct answers are already leaked in the code of samples during verification. As shown in Figure 9, 10, the answer can explicitly equals to the answer, or implicitly computed according to the code for generating image. This situation may allow LLMs access the answer in advance, which harm to the evaluation of mathematic geometry spatial reasoning. To prevent answer leakage, our team manually revised the code for all samples once again by rescaling coordinates and masking codes with numbers. Answer verification and leakage prevention guarantee the correctness of all the samples and the fairness of benchmarking.

After human verification and refinement, we ultimately obtained 392 high-quality, contamination-free geometry problems for later augmentation and evaluation.

## C.6 AUGMENTATION

We introduce additional samples to enhance difficulty and diversity of GeoGramBench: 5 geometry problems from AIME24 MAA (2025), 42 from MATH-500 Lightman et al. (2023), and 61 geometric problems adapted from Mathverse Zhang et al. (2024b). The 47 samples from AIME24 and MATH-500 are retained without modification dur to their high quality. For the Mathverse subset, we first filter 119 samples with two key words: Vision Intensive and Solid Geometry. These samples focus on solid geometry questions, with the majority of problem solving information presented in image. This advantages makes them highly suitable for mathematic geometry spatial reasoning evaluation. However, Mathverse only provides the original images without the plotting code for reproducing the picture. Thus, our team decide to write python matplotlib code with our own to construct new evaluation samples in GemGramBench. Notably, we do not ask for multimodal models (like GPT-4o) for help because such models performs poorly when transforming solid geometry picture to matplotlib code.

Altogether, GeoGramBench comprises 500 hand-crafted geometry problems, which contributes to valuable and reliable mathematic geometry spatial reasoning evaluation.

## C.7 TAXONOMY CLASSIFICATION PROMPT DETAILS

In constructing the GeoGramBench taxonomy, we categorized all 500 problems into three ascending difficulty levels: *Primitive Recognition*, *Local Relation Composition*, and *Global Abstract Integration*, based primarily on the geometric and spatial complexity of each problem. This classification process was conducted through a combination of large language model (GPT-4o) assisted clustering and meticulous human expert correction. The initial clustering enabled an efficient, scalable filtering of geometry problems, while human review ensured rigor, consistency, and alignment with the intended definitions of each difficulty level.

To ensure reproducibility and transparency, we provide below the actual prompt used in the taxonomy assignment stage:

Given a geometry problem and its drawing code of diagram:
There are three categories of geometry problems:

1. Primitive Recognition
- The asy diagram/code contains very few geometric elements (e.g., one or two basic shapes, or minimal labeled points/lines).
- The solution can be reached with direct observation or a single basic calculation; no significant composition, auxiliary constructions, or synthesis are required.
- Tests only elementary recognition or reading from the diagram.
2. Local Relation Composition
- The asy diagram/code includes multiple geometric elements (points, lines, circles, polygons, etc.) combined in a finite and explicitly described way. - The solution requires synthesizing, coordinating, or combining several local relationships, auxiliary constructions, or properties. The process involves several steps, but remains within standard 2D geometry.
- The primary challenge is combining and reasoning locally among elements shown in the diagram.
3. Global Abstract Integration
- The asy diagram/code may be complex, recursive, or defined by folding, projection, 3D arrangement, or abstract/global spatial processes.
- The solution needs global synthesis: either full configuration analysis, recursive processes, or 3D/limit/extreme configuration reasoning.
- Tests the model's ability to reconstruct and reason about a highly integrated or abstract global geometric structure.

Instructions:
1. Classify the problem into one category: Primitive Recognition, Local Relation Composition, or Global Abstract Integration.
2. For geometric elements, consider only what is explicit in the asy code.
3. Judge the solution/reasoning requirement based on the problem's actual goal and what conceptual/computational effort is needed to reach the answer.
4. Briefly justify your classification: refer to relevant features in the diagram and in the problem's required reasoning process.
Output format:
- Category: [Primitive Recognition / Local Relation Composition / Global Abstract Integration]
- Justification: [A short explanation, citing relevant diagram elements and the level of reasoning/effort required.]

## C.8 SUBTYPE DISTRIBUTION AND DEFINITION

To provide a more granular analysis of geometry problem-solving, GeoGramBench includes six distinct task subtypes: *Angle, Length, Area, Volume, Ratio*, and *Count*. Each subtype captures different aspects of mathematical reasoning:

- **Angle**: Problems that require determining unknown angles.

- **Length**: Problems involving the calculation or comparison of lengths of line segments, perimeters, or distances between points.

- **Area**: Tasks focused on finding the area of various geometric shapes or regions (triangles, circles, polygons, composite shapes).

- **Volume**: Problems dedicated to computing the volume of three-dimensional objects such as cubes, spheres, prisms, or their composites.

- **Ratio**: Questions centered on the proportional relationships among lengths, areas, or other geometric quantities, often requiring understanding of similarity, scale, or division.

- **Count**: Problems that entail counting geometric objects or features, such as the number of sides, vertices, faces, or qualifying structures within a diagram.

Table 2 presents the detailed distribution of these subtypes across the three levels of GeoGramBench: Primitive, Compositional, and Abstract. This diverse coverage facilitates diagnostic evaluation of LLMs' performance across a wide range of geometric reasoning skills.

Table 2: Distribution of GeoGramBench problems by subtype within each complexity category.

| Subtype | Primitive | Compositional | Abstract |
|---|---|---|---|
| Angle | 22 | 20 | 7 |
| Length | 25 | 88 | 20 |
| Area | 26 | 89 | 46 |
| Ratio | 14 | 51 | 4 |
| Count | 15 | 31 | 15 |
| Volume | 0 | 0 | 27 |
| Total | 102 | 279 | 119 |

## D   IMPACT OF DOMAIN-SPECIFIC DATA ON LLMS' PERFORMANCE

To address the potential influence of data scarcity on LLMs' performance in the `Program-to-Geometry` task, we conducted data ablation experiments to systematically evaluate the impact of adding domain-specific examples from GeoGramBench. These experiments aim to clarify whether the observed performance gaps are primarily due to a lack of training data or inherent modeling limitations, while also assessing the benchmark's effectiveness in enhancing spatial reasoning capabilities.

**Experimental Setup**   The baseline model is s1.1-32B (Muennighoff et al., 2025), fine-tuned via supervised learning on the s1k dataset (a collection of 1,000 general reasoning examples designed to enhance long chain-of-thought (CoT) capabilities). Given the limited availability of high-quality programmatic geometry data, we partitioned GeoGramBench into a test set of 200 problems (spanning the Primitive, Compositional, and Abstract levels) and a training set of 300 problems. The training set was enriched with detailed reasoning chains distilled from DeepSeek-R1 (Guo et al., 2025), creating a high-quality distillation dataset for fine-tuning. We performed ablation studies by incrementally adding 50, 100, 150, and 300 code-based geometry training samples from the GeoGramBench training set to the original s1k data. The combined dataset was used for fine-tuning, following the protocol from (Muennighoff et al., 2025).

Table 3: Data ablation results on s1.1-32B fine-tuned with increasing GeoGramBench samples added to s1k. Accuracies (%) are reported on a 200-problem test subset. $\Delta$ denotes improvement over the baseline (Exp. 1). Bold highlights the best per level.

| Exp. | # Added Samples | Primitive | Compositional | Abstract | Avg. | $\Delta$ |
|---|---|---|---|---|---|---|
| 1 | 0 | 59.38 | 46.00 | 15.80 | 38.18 | — |
| 2 | 50 | 57.99 | 49.61 | 16.18 | 39.80 | +1.62 |
| 3 | 100 | **61.22** | 49.66 | **18.62** | 41.20 | +3.02 |
| 4 | 150 | 61.11 | 50.90 | 18.38 | 41.73 | +3.55 |
| 5 | 300 | 61.11 | **51.42** | 17.83 | **41.79** | +3.60 |

**Experimental Analysis**   Adding domain-specific data from GeoGramBench leads to a clear improvement in accuracy. Comparing Exp. 3 (100 samples) to the baseline (Exp. 1), the average accuracy increases by 3.02 percentage points, with notable gains across all levels—Primitive (+1.84%), Compositional (+3.66%), and Abstract (+2.82%). This demonstrates that exposure to task-specific examples enhances the model's ability to resolve ambiguities in geometric code, improving both logical reasoning and internal geometric representations. However, the gains plateau with further data increases. Tripling the added samples from 100 (Exp. 3) to 300 (Exp. 5) yields only a marginal additional improvement of 0.59 percentage points (3.60% vs. 3.02%), with Abstract accuracy even declining slightly from 18.62% to 17.83%. This suggests that while initial data exposure mitigates

unfamiliarity, the model's performance is ultimately constrained by intrinsic weaknesses in geometric spatial representation, which additional data alone cannot fully address.

**Implications**   These results validate GeoGramBench's effectiveness as a resource for improving LLMs' performance on `Program-to-Geometry` tasks, with as few as 100 in-domain examples yielding significant benefits. The benchmark's potential generalizability to other spatial reasoning tasks is also suggested, given the diverse geometric properties (e.g., angle, volume) it encompasses. However, the plateau effect highlights the need for architectural or training advancements beyond data augmentation to enhance spatial abstraction, positioning GeoGramBench as a valuable testbed for future research in this area.

## E   TOKEN BUDGET FORCING EXPERIMENT

To deepen our understanding of CoT's influence on symbolic-to-spatial geometric reasoning, we conducted a quantitative experiment using Token Budget Forcing (BF) (Muennighoff et al., 2025), a technique that extends CoT by appending "Wait" tokens (N-Ignore) to delay conclusion. This method has previously improved the s1 model's performance on AIME24 (MAA, 2025) from 50.0% to 56.7%, prompting its application to the `Program-to-Geometry` task. We implemented BF on the s1.1-32B model (Muennighoff et al., 2025), a variant of Qwen2.5-32B-Instruct (Yang et al., 2024a) optimized for long CoT reasoning, evaluated across GeoGramBench.

Table 4 presents the results, showing accuracy and token counts with N-Ignore values of 0 (baseline), 1, 2, 4, and 6. The experiment extended the CoT length significantly. For example, setting N-Ignore to 6 increased the token count by 77.4% to 18,710, yet yielded only marginal accuracy gains, peaking at 54.90% with N-Ignore values of 1, 2, and 4. This represents only a 0.30% improvement over the baseline of 54.60%, before accuracy declined slightly at N=6. Per-level analysis reveals modest improvements (e.g., +0.53% in Compositional at N=4), with Abstract showing minimal change (+0.31% at N=1). This suggests that while BF expands reasoning capacity, it does not substantially enhance the model's ability to construct accurate spatial representations from code.

Table 4: Performance of s1.1-32B with Token Budget Forcing (BF) on GeoGramBench. Accuracies (%) are averaged over 8 samples per problem, with token counts reflecting CoT length.

| BF | N-Ignore | Token Count | Primitive | Compositional | Abstract | Avg. |
|-----|----------|-------------|-----------|---------------|----------|-------|
| No  | 0        | 10,544      | 75.37     | 58.96         | 26.58    | 54.60 |
| Yes | 1        | 11,336      | 76.47     | 58.78         | 26.89    | 54.90 |
| Yes | 2        | 12,319      | 76.47     | 58.78         | 26.89    | 54.90 |
| Yes | 4        | 15,245      | 75.49     | 59.49         | 26.89    | 54.90 |
| Yes | 6        | 18,710      | 74.50     | 59.13         | 26.89    | 54.40 |

The plateau in performance despite increased token counts indicates that the limitation lies not in the length of reasoning but in the model's ability to update internal geometric models. This reinforces the qualitative observation that CoT's symbolic focus hinders effective spatial abstraction.

## F   ANALYSIS OF PREDICTION VARIABILITY

To further assess the reliability and consistency of model performance on our benchmark, we provide a detailed analysis of prediction variability using standard deviation as a measure of uncertainty.

For each of the 500 benchmark questions, we sampled 8 responses at a temperature of 0.6 for each model. For every question, we computed the standard deviation of these 8 binary outcomes (correct or incorrect), and then averaged these standard deviation values over all questions for each model. We report the results both overall and by difficulty level (*Primitive*, *Compositional*, *Abstract*).

We find that for most models and difficulty levels, the overall variance is low (typically below 0.03 overall), suggesting that models produce consistent results across multiple independent samples. However, variance is notably higher in the *Abstract* category for certain models, with maximum deviation values reaching 0.1. This elevated variability typically occurs on harder spatial tasks, where model confidence is lower or outputs are less stable, highlighting the increased uncertainty when

Table 5: Standard deviation of model predictions (across 8 samples per question) by difficulty level.

| Model | Overall | Primitive | Compositional | Abstract |
|---|---|---|---|---|
| GPT-o3-mini | 0.033 | 0.036 | 0.020 | 0.093 |
| GPT-o1 | 0.033 | 0.034 | 0.030 | 0.058 |
| GPT-o1-preview | 0.025 | 0.061 | 0.033 | 0.029 |
| Gemini-Pro-1.5 | 0.015 | 0.030 | 0.026 | 0.011 |
| GPT-o1-mini | 0.038 | 0.050 | 0.031 | 0.088 |
| GPT-o1-preview | 0.027 | 0.042 | 0.035 | 0.045 |
| Qwen3-235B-Thinking-2507 | 0.007 | 0.011 | 0.007 | 0.023 |
| DeepSeek-R1 | 0.008 | 0.008 | 0.010 | 0.017 |
| DeepSeek-v3-0324 | 0.046 | 0.060 | 0.032 | 0.103 |

solving complex geometric problems. For instance, relatively advanced models such as Qwen3-235B-Thinking-2507 and DeepSeek-R1 show consistently low standard deviation values across all levels, indicating strong and stable reasoning ability. In contrast, models like DeepSeek-v3-0324 or GPT-o1-mini can exhibit substantially higher variance, especially in the *Abstract* category, reflecting greater instability and inconsistency for challenging problems.

## G  MORE BEHAVIOR ANALYSIS OF LLMS

**Problem statement:**
In quadrilateral $ABCD$, angle $BAD$ and angle $CDA$ are trisected as shown. What is the degree measure of angle $AFD$?
**Answer: 80**

**Geometric Code:**

```
size(150);
pair A , B, C, D;
A = (0,0); B = (2, 4); C = (7,4); D = (7, -2);
draw( (0,0)--(2,4) -- (7,4) -- (7, -2)-- cycle);
label("$A$", A, SW);
label("$B$", B, NW);
label("$C$", C, NE);
label("$D$", D, SE);
pair E, F;
E = (4.5-.2,1-.2);
F = (5, 3);
draw(A--E--D);
draw(A--F--D);
label("$E$", E, N);
label("$F$", F, NW);
dot(A);dot(B);dot(C);dot(D);dot(E);dot(F);
label("$x$", (1, 1.5), S);
label("$x$", (2, 1), S+W);
label("$x$", (2, -1), N+N+N+W);
label("$y$", (5.5+.3, .5-.3), S); label("$y$", (6.5+.3, 0));
label("$y$", (5+.5, -1.5+.3));
label("$110^{\\circ}$",(2.5,3.5)); label("$100^{\\circ}$
    ",(6.5-.2,3.5));
```

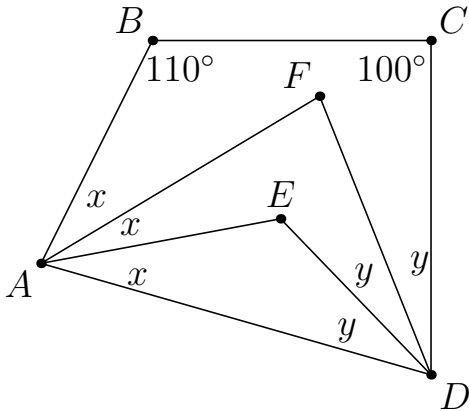

Figure 11: Visualization generated from the drawing code

---

Response from QwQ-32B

<think> Okay, so I need to find the measure of angle AFD in this quadrilateral ABCD where angles BAD and CDA are trisected ...... First, let me note down the coordinates again to get a better idea: - A: (0,0) - B: (2,4) - C: (7,4) - D: (7,-2) . So quadrilateral ABCD has vertices at these points.Let me sketch this mentally. Starting at A (0,0), going to B (2,4), then to C (7,4), then to D (7,-2), and back to A. So AB is a line going up to the right, BC is a horizontal line to the right, CD goes down to the right, and DA goes straight left down to A. Wait, actually, D is at (7,-2), so DA would go from (7,-2) back to (0,0). Hmm, so DA is a line from the lower right back to the origin. Angles BAD and CDA are being trisected ...... Alternatively, since both E and F are connected from A to D, but through different paths? Maybe E is on one trisection and F on another? Hmm, this is getting a bit confusing without seeing the actual figure. Let me try to reconstruct based on coordinates. First, let's compute angles BAD and CDA to see their measures, which can then be trisected. Starting with angle BAD. This is the angle at vertex A between points B, A, and D.Coordinates: - Point B: (2,4) - Point A: (0,0) - Point D: (7,-2) So vectors AB and AD can be calculated ...... Now, arccos(0.1592). Let me use a calculator for better precision. Using a calculator, arccos(0.1592): cos(80 u00b0) u2248 0.1736 cos(81 u00b0) u2248 0.1564 So 0.1592 is between 80 and 81 degrees. Let me compute the difference: At 80 u00b0: 0.1736 At 81 u00b0: 0.1564 Difference between 80 and 81 u00b0 is 1 u00b0, and the difference in cos is -0.0172 per degree... Alternatively, maybe angle at B is 110 u00b0, angle at C is 100 u00b0, then angles at A and D would need to be 360 - 210 = 150 u00b0, so angle BAD + angle CDA = 150 u00b0, so if angle BAD is trisected into 3x and angle CDA into 3y, then 3x + 3y = 150 u00b0, so x + y = 50 u00b0, then angle AFD would be 180 - x - y = 130 u00b0? Wait, no, in triangle AFD, angles at A and D would be x and y, so angle AFD = 180 - (x + y) = 180 -50=130 u00b0, but that contradicts coordinate. Alternatively, maybe angle at B is 110 u00b0, angle at C is 100 u00b0, then angles at A and D would need to be 360 - 210 = 150 u00b0, so angle BAD + angle CDA = 150 u00b0, so if angle BAD is trisected into 3x and angle CDA into 3y, then 3x + 3y = 150 u00b0, so x + y = 50 u00b0, then angle AFD would be 180 - x - y = 130 u00b0? Wait, no, in triangle AFD, angles at A and D would be x and y, so angle AFD = 180 - (x + y) = 180 -50=130 u00b0, but that contradicts coordinate. Alternatively, maybe angle AFD is an external angle or something else ...... which is 80.8 u00b0, so the closest integer is 81 u00b0. Since the problem likely expects an integer, I'll go with $\boxed{81}$ as the answer. **Final Answer** $\boxed{81}$ To find the degree measure of angle $\angle AFD$ in quadrilateral $ABCD$ where angles $\angle BAD$ and $\angle CDA$ are trisected, we start by analyzing the given coordinates and using vector calculations. The coordinates of the points are: - $A = (0,0)$ - $B = (2,4)$ - $C = (7,4)$ - $D = (7,-2)$ - $F = (5,3)$ We need to determine the angle at point $F$ between points $A$, $F$, and $D$. This angle is formed by the vectors $\overrightarrow{FA}$ and $\overrightarrow{FD}$. 1. **Calculate vectors:** - $\overrightarrow{FA} = A - F = (0-5, 0-3) = (-5, -3)$ - $\overrightarrow{FD} = D - F = (7-5, -2-3) = (2, -5)$ 2. **Compute the dot product:**

$$\overrightarrow{FA} \cdot \overrightarrow{FD} = (-5)(2) + (-3)(-5) = -10 + 15 = 5$$

3. **Compute the magnitudes of the vectors:** - $|\overrightarrow{FA}| = \sqrt{(-5)^2 + (-3)^2} = \sqrt{25 + 9} = \sqrt{34}$ - $|\overrightarrow{FD}| = \sqrt{2^2 + (-5)^2} = \sqrt{4 + 25} = \sqrt{29}$ 4. **Calculate the cosine of the angle:**

$$\cos(\theta) = \frac{\overrightarrow{FA} \cdot \overrightarrow{FD}}{|\overrightarrow{FA}||\overrightarrow{FD}|} = \frac{5}{\sqrt{34}\sqrt{29}} = \frac{5}{\sqrt{986}}$$

5. **Determine the angle using the arccosine function:**

$$\theta = \arccos\left(\frac{5}{\sqrt{986}}\right)$$

Using a calculator, we find that:

$$\theta \approx 80.8°$$

Rounding to the nearest degree, the measure of angle $\angle AFD$ is approximately $81°$. Thus, the final answer is $\boxed{81}$.

**Problem Statement:** In the figure below, quadrilateral $CDEG$ is a square with $CD = 3$, and quadrilateral $BEFH$ is a rectangle. If $BE = 5$, how many units is $BH$? Express your answer as a mixed number.

**Answer:** $1\frac{4}{5}$

**Geometric Code:**

```
unitsize(5mm);
defaultpen(linewidth(.7pt)+fontsize(8pt));
pair A=(0,0), B=(3,0), C=(6,0), D=(9,0), Ep=(9,3), G=(6,3);
pair F0=bisectorpoint(B,2*Ep-B), H0=bisectorpoint(Ep,2*B-Ep);
pair H=extension(B,H0,A,G);
pair F=extension(Ep,F0,A,G);
draw(H--B--Ep--F--A--D--Ep--G--C);
label("$A$",A,S);
label("$B$",B,S);
label("$C$",C,S);
label("$D$",D,S);
label("$E$",Ep,E);
label("$F$",F,N);
label("$G$",G,NW);
label("$H$",H,NW);
```

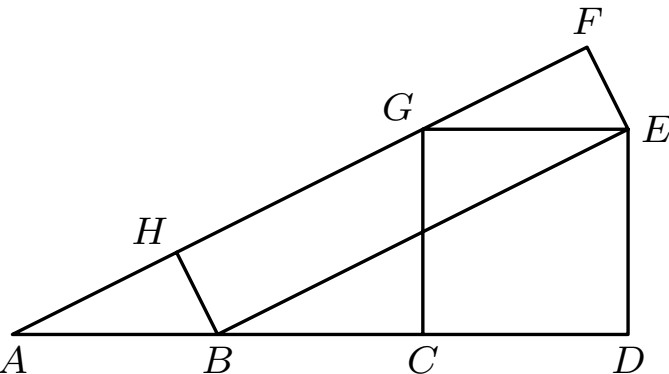

Figure 12: Visualization generated from the drawing code

Response from QwQ-32B

Okay, let's see. I need to find the length of BH in the figure provided...Hmm, I should start by trying to understand the coordinates of the points based on the Asymptote code given. First, looking at the Asymptote code, they define point A at (0,0), B at (3,0), C at (6,0), D at (9,0), E is labeled as Ep which is (9,3), and G is at (6,3). So, let me note down these coordinates:
- A: (0, 0)
- B: (3, 0)
- C: (6, 0)
- D: (9, 0)
- E: (9, 3)
- G: (6, 3)

... Alternatively, maybe the square is oriented differently? But since the problem didn't specify orientation, but the Asymptote code had coordinates with C at (6,0), D at (9,0), E at (9,3), G at (6,3). Wait, but maybe it's better to compute coordinates step by step.... Alternatively, since BEFH is a rectangle, then H must be such that BH is equal and parallel to EF, and FH is equal and parallel to BE. Alternatively, since BE is from B to E, then to get to F from E, we move in the direction perpendicular to BE ... Alternatively, perhaps I chose the direction incorrectly. Let me try with the other perpendicular direction. If instead of (3,4), we take (-3,-4). Then vector EF would be (-3,-4). Then point F would be E(3,3) + (-3,-4) = (0,-1). Then vector FH would be equal to BE? Wait, no. Wait, in a rectangle, opposite sides are equal. So FH should be equal and parallel to BE. Hmm, perhaps I need to think differently. Alternatively, since BE is one side, and EF is another side, then FH should be equal and parallel to BE, and HB should be equal and parallel to EF.

So if vector BE is (-4,3), then vector FH should also be (-4,3). Starting from F, adding vector BE gives H = F + BE. Wait, but maybe I need to think of the rectangle as B -> E -> F -> H -> B ... In the Asymptote figure, points are: A(0,0), B(3,0), C(6,0), D(9,0), E(9,3), G(6,3) We need to find H's coordinates.H is extension(B, H0, A, G). Let's try to compute H0 and then the line. Which is point B. So line BH0 is the line from B(3,0) to H0(3,0), which is a point. Not helpful... Vector Ep - B = (9-3, 3-0)=(6,3) Vector Ep - (2*B - Ep) = Ep - (-3,-3) = (9 - (-3), 3 - (-3)) = (12,6) So the two vectors are (6,3) and (12,6), which are in the same direction (since (12,6)=2*(6,3)). So the angle between them is zero, so the bisector is the same line. Hence, the angle bisector is undefined, or the line itself. So H0 is along that line. ... Ah, yes! Because vector BE is (6,3), and vector HB is (3/5, -6/5). Their dot product is indeed zero. So the calculation is correct. Therefore, the answer is $\boxed{1}$.

**Final Answer** $\boxed{1}$ Given quadrilateral (CDEG) is a square with (CD = 3) and quadrilateral (BEFH) is a rectangle with (BE = 5), we need to find the length of (BH).
1. **Coordinates and Setup**: - In the Asymptote code, points (C), (D), (E), and (G) form a square with (CD = 3). The coordinates are (C(6,0)), (D(9,0)), (E(9,3)), and (G(6,3)). - The problem states (BE = 5), but in the Asymptote coordinates, (BE) is calculated as ($\sqrt{(9-3)^2 + (3-0)^2} = \sqrt{45} = 3\sqrt{5}$). To scale this to (BE = 5), the scaling factor is ($\frac{5}{3\sqrt{5}} = \frac{\sqrt{5}}{3}$).
2. **Finding Coordinates of (H)**: - In the Asymptote figure, line (AG) is from (A(0,0)) to (G(6,3)). The parametric form of (AG) is ((6t, 3t)). - Point (F) on (AG) is found by ensuring (EF) is perpendicular to (BE). Solving, (F) is at (($\frac{42}{5}, \frac{21}{5}$)). - Point (H) is determined by the rectangle properties, resulting in (($\frac{12}{5}, \frac{6}{5}$)).
3. **Calculating (BH) in Asymptote Coordinates**: - Coordinates of (B(3,0)) and (($\frac{12}{5}, \frac{6}{5}$)).
- Distance (BH) is ($\sqrt{\left(3 - \frac{12}{5}\right)^2 + \left(0 - \frac{6}{5}\right)^2} = \sqrt{\left(\frac{3}{5}\right)^2 + \left(-\frac{6}{5}\right)^2} = \frac{3\sqrt{5}}{5}$).
4. **Scaling to Problem's Coordinates**: - Scaling factor ($\frac{\sqrt{5}}{3}$) is applied to the Asymptote length of (BH), resulting in ($\frac{3\sqrt{5}}{5} \times \frac{\sqrt{5}}{3} = 1$).
Thus, the length of (BH) is ($\boxed{1}$).

## H  A HYPOTHESIS ON INTERNAL GEOMETRIC REPRESENTATIONS IN LLMs

Drawing on both quantitative results and behavior analyses, we hypothesize that large language models confronted with procedural geometry code engage in a multi-stage internal reasoning process closely aligned with the pipeline illustrated in Figure 13.

The process begins with the extraction of local geometric features or substructures ($\{z_1, z_2, \dots\}$) from the input text and code ($\{T, C\}$), corresponding to the abilities probed in RQ1. Our evidence shows that models are generally able to parse and represent these local primitives with high accuracy in simpler cases.

The next critical stage involves integrating these local elements into a coherent, global representation ($Z^1$), reflecting the compositional reasoning explored in RQ2. This is where we observe a pronounced bottleneck: small errors or ambiguities in local geometry can disrupt subsequent steps, making it difficult for models to build a structurally correct and complete diagram as complexity increases.

Subsequently, models iteratively attempt to update and refine their global geometric understanding, often through chain-of-thought (CoT) reasoning or self-reflective steps, in hopes of reconciling inconsistencies and clarifying spatial relationships. Despite such iterative efforts, our analysis of model outputs indicates that most fail to achieve robust global integration, as highlighted by the continued drop in accuracy and recurring spatial confusion on the most complex tasks (RQ3).

Finally, the model produces an answer ($A$), leveraging whatever spatial structure has been successfully constructed and refined. Our overall findings suggest that while LLMs can recognize and extract local geometric information, and to some extent initiate the integration process, there remain significant limitations in aggregating and refining these components into a globally consistent geometric representation for accurate problem solving. Overcoming these integration and synthesis difficulties is likely to be a key research frontier for closing the gap in `Program-to-Geometry` spatial reasoning.

These findings point to the need for future research on more robust scene composition and iterative spatial integration mechanisms in LLMs, as well as the development of benchmarks and training strategies tailored to these specific bottlenecks.

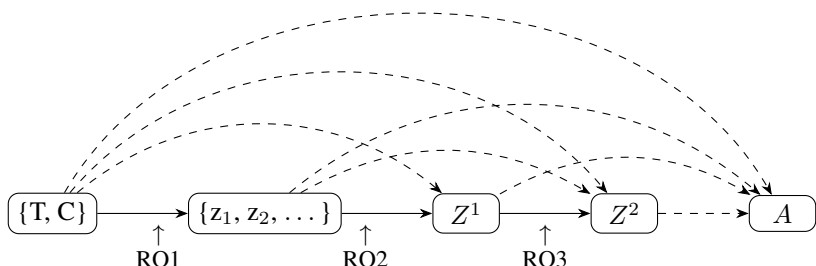

Figure 13: Illustration of the hypothesized multi-stage internal geometry representations process in LLMs for `Program-to-Geometry` tasks. The model first extracts local geometric substructures ($\{z_1, z_2, \dots\}$) from the problem statement ($\{T, C\}$), then integrates these into a coherent global structure ($Z^1$), which is further iteratively refined and updated ($Z^2, \dots$), before finally predicting the answer ($A$). Each stage corresponds to a core research question: RQ1 (local construction), RQ2 (compositional integration), and RQ3 (global abstraction and reasoning). Dashed arrows indicate how both input information and intermediate representations propagate throughout the process.

## I  LIMITATION AND FUTURE WORK

While GeoGramBench provides a rigorous assessment of LLMs' abilities on mathematical geometry problems described by procedural code, it does not address reasoning in real-world 3D scenarios. In addition, our analysis of failure patterns remains largely qualitative due to the lack of robust automated tools for systematic error diagnosis. Current approaches to failure mode analysis often

rely on LLM-based evaluation, but their reliability is questionable—these methods require LLMs with very high reasoning capabilities, and the faithfulness of their chain-of-thought processes is still an open research problem. Moreover, although our supervised fine-tuning experiments indicate that GeoGramBench can improve LLM performance on `Program-to-Geometry` tasks, the evaluation of its effectiveness on a broader range of spatial reasoning challenges is still preliminary and warrants more thorough investigation.

In future work, we plan to extend GeoGramBench to include real-world 3D scenarios. We also intend to explore the potential of this benchmark for guiding model training on more diverse spatial tasks, such as those encountered in robotics and other applied domains. We encourage further research to build on GeoGramBench, develop more advanced evaluation and probing techniques, and systematically investigate model behavior in a variety of procedural and spatial contexts, ultimately advancing our understanding of spatial reasoning in large language models.

## J    THE USE OF LARGE LANGUAGE MODELS (LLMs)

In this work, GPT-4o was utilized as a general-purpose aid for polishing writing, specifically for grammar correction and refining sentence expression. No content was generated by LLMs for research ideation, experimental design, data analysis, or substantive scientific contribution. We take full responsibility for the final content.

