# OpenReview forum: "GeoGramBench: Benchmarking the Geometric Program Reasoning in Modern LLMs"
_ICLR.cc/2026/Conference — ICLR 2026 Poster_

### Official Review · Reviewer_Dim1 · 2025-10-30

**Soundness:** 3
**Presentation:** 3
**Contribution:** 3
**Rating:** 6
**Confidence:** 4

**Summary:**

How well can models engage in geometric reasoning? How well can models engage jointly over structured symbolic and spatial representations? The authors formalize a new evaluation task (Program-to-Geometry), collate a new testbed for this task, and apply that testbed to assess current models.

**Strengths:**

The paper is well-motivated and engages with an interesting reasoning challenge (understanding geometric reasoning as it relates to symbolic representations of those problems). The authors' benchmark is quite well-structured in my opinion --- particularly in having multiple difficulty levels and coding by concept. That kind of granular decomposition will surely help other researchers and model developers better isolate and probe model capacities!

I also appreciated the diversity of model analyses the authors engaged in (both quantitative and more qualitative, e.g., Section 6).

I also found the authors' formalization and explication of two kinds of answer leakage very important and a valuable contribution in its own right!

**Weaknesses:**

While the empirical work is wide-ranging, I think the paper could be improved in clarity and rigour of the empirical analyses. For instance, the authors sample with a moderately high temp (0.6), to my understanding. However, no error bars are reported? How variable are the results? It would be good for the authors to present some sense of performance variation across tasks (e.g., 95% CIs around the mean; stdev, etc).

I also found the phrasing of Program-to-Geometry in Section 3.1 somewhat confusing. It seems like there are two capabilities the authors are interested in (as they lay out in that 3.1). But is Program-to-Geometry as a task both? Just one? It's worth clarifying (and I found that confusing...)

It also would be good for the authors to clarify how their work relates to https://arxiv.org/pdf/2408.08313, which seems quite similar, especially related to RQ1 (which was at ICLR last year and is not cited).

**Questions:**

- How much variance is there in model performance (see question above, re: error bars)?
- For future work, not this revision :) have the authors looked at reasoning token usage across the difficulty levels in the benchmark?

---

> ### Author Response · Authors · 2025-11-23
> **Author Response to Reviewer Dim1 (Part 1/2)**
>
> Thank you for your thoughtful review and constructive comments. We would appreciate any additional feedback to help us further improve the work.
>
> ---
>
> >**Q1**. While the empirical work is wide-ranging, I think the paper could be improved in clarity and rigour of the empirical analyses. For instance, the authors sample with a moderately high temp (0.6), to my understanding. However, no error bars are reported? How variable are the results? It would be good for the authors to present some sense of performance variation across tasks (e.g., 95% CIs around the mean; stdev, etc).
>
> > How much variance is there in model performance (see question above, re: error bars)?
>
> ---
>
> **A1**: Thank you for your helpful feedback regarding the reporting of error bars and performance variability. We agree that presenting measures of variation such as standard deviation offers important insight into the reliability and consistency of empirical results.
>
> To address this, we have calculated and summarized the standard deviation of model performance across all levels. For each of the 500 benchmark questions, we sampled 8 responses per model (with temperature = 0.6). For each question, we computed the standard deviation over the 8 binary outcomes (correct/incorrect), and then averaged these values over all questions for each model, both overall and by difficulty level.
>
> The results are shown in the table below:
>
> | Model                       | Overall  | Primitive  | Compositional | Abstract |
> |-----------------------------|----------|----------|-------------|----------|
> | GPT-o3-mini                 | 0.033    | 0.036    | 0.020       | 0.093    |
> | GPT-o1                      | 0.033    | 0.034    | 0.030       | 0.058    |
> | GPT-o1-preview              | 0.025    | 0.061    | 0.033       | 0.029    |
> | Gemini-Pro-1.5              | 0.015    | 0.030    | 0.026       | 0.011    |
> | GPT-o1-mini                     | 0.038    | 0.050    | 0.031       | 0.088    |
> | GPT-o1-preview                  | 0.027    | 0.042    | 0.035       | 0.045    |
> | Qwen3-235B-Thinking-2507         | 0.007    | 0.011    | 0.007       | 0.023    |
> | DeepSeek-R1                 | 0.008    | 0.008    | 0.010       | 0.017    |
> | DeepSeek-v3-0324            | 0.046    | 0.060    | 0.032       | 0.103    |
>
> These values suggest that, for most models and difficulty levels, the variance in sampled predictions is fairly low (typically <0.03 overall), indicating consistent performance across multiple samples. Higher variance values are observed in specific models and especially in the Abstract category (up to 0.1), which often correspond to harder tasks and situations where models are less confident or predictions are more unstable.
>
> We believe this analysis provides additional clarity regarding reliability and stability of the reported results. We have included this information and clarified our approach to quantifying performance variability in the revision.
>
>
> > **Q2**. I also found the phrasing of Program-to-Geometry in Section 3.1 somewhat confusing. It seems like there are two capabilities the authors are interested in (as they lay out in that 3.1). But is Program-to-Geometry as a task both? Just one? It's worth clarifying (and I found that confusing...)
>
>
> **A2**: Thank you for pointing out the potential confusion in the phrasing of Program-to-Geometry in Section 3.1. To clarify, Program-to-Geometry as a task encompasses both capabilities mentioned: (a) the ability to accurately construct geometric diagrams from symbolic code, and (b) the ability to perform spatial reasoning and mathematical problem solving based on these constructed diagrams. These abilities are sequential and interdependent—the model must first correctly interpret and build the geometric figure from the code, and then use this representation for reasoning to solve the math question.
>
> Our experimental results further illustrate this relationship. On simpler problems (Primitive level), top models achieve over 80% accuracy, demonstrating strong performance in both diagram construction and reasoning. However, as diagram complexity increases (Compositional and Abstract levels), accuracy drops, falling below 50% in the most complex cases. Qualitative analysis shows that as models struggle to construct accurate geometric figures, their ability to reason and solve problems based on these figures is also limited.
>
> In summary, Program-to-Geometry is designed as a comprehensive assessment of both skills, and success in the task requires proficiency in both constructing geometric representations and reasoning over them.

---

> ### Author Response · Authors · 2025-11-23
> **Author Response to Reviewer Dim1 (Part 2/2)**
>
> > **Q3**. It also would be good for the authors to clarify how their work relates to https://arxiv.org/pdf/2408.08313, which seems quite similar, especially related to RQ1 (which was at ICLR last year and is not cited).
>
> **A3**: Thank you for your question regarding the relationship between our work and SGP-Bench (ICLR 2025). SGP-Bench is indeed closely related, especially to our RQ1 about LLMs' ability to understand and represent basic geometric elements from program code.
>
> Both benchmarks focus on the symbolic-to-spatial reasoning abilities of LLMs, but the motivation and task design differ. SGP-Bench primarily evaluates perception tasks, such as recognizing shape, color, and count, through questions about patterns and visual invariance. In contrast, GeoGramBench is designed to assess models' abilities in geometric abstraction and mathematical reasoning. Our task requires LLMs not only to perceive and reconstruct geometric figures from symbolic code, but also to solve math problems using the internal geometric representations, which demands deeper spatial logic and abstraction. SGP-Bench mainly uses multiple-choice questions, suitable for perception benchmarks, while GeoGramBench adopts numeric answers to rigorously test mathematical reasoning abilities.
>
>
> We have cited SGP-Bench and added a dedicated discussion to the Related Works section in our revised manuscript.
>
>
> > **Q4**. For future work, not this revision :) have the authors looked at reasoning token usage across the difficulty levels in the benchmark?
>
>
> **A4**: Thank you for this interesting suggestion regarding reasoning token usage across different difficulty levels in our benchmark. Below, we present the reasoning token counts for several models of varying parameter size:
>
> | Model                        | Primitive | Compositional | Abstract |
> |-----------------------------|-----------|---------------|----------|
> | DeepSeek-Distill-Qwen-32B   |   5539    |     8840      |   9333   |
> | DeepSeek-Distill-Qwen-7B    |   7100    |    10357      |  12743   |
> | DeepSeek-Distill-Qwen-1.5B  |   7723    |    13457      |  14192   |
>
> 1. For all models, the number of reasoning tokens increases as the geometric complexity of the problem rises. This shows that more complex problems require longer chains of thought for models to interpret code, construct figures, and process their reasoning and reflections. As expected, the Abstract category always has the highest token usage compared to the Primitive category.
>
> 2. Models with larger parameter sizes require fewer reasoning tokens for problems of the same difficulty. For example, DeepSeek-Distill-Qwen-32B consistently uses fewer tokens across all levels compared to the 7B and 1.5B variants, suggesting that larger models, with stronger representation and reasoning capabilities, need less correction and reflection during problem solving.
>
> These results provide useful insights into both how geometric complexity and model size affect the reasoning process.

---

> > ### Comment · Reviewer_Dim1 · 2025-11-24
> > **Strong response**
> >
> > Thank you for the extremely thorough set of responses! I have boosted my score accordingly and think this paper should be accepted.

---

> > > ### Author Response · Authors · 2025-11-25
> > > **Author Response to Reviewer Dim1**
> > >
> > > We are grateful for your thoughtful feedback and the favorable score adjustment, which have been highly encouraging to us. Thank you for your support and recognition!

---

### Official Review · Reviewer_XpCL · 2025-10-31

**Soundness:** 2
**Presentation:** 3
**Contribution:** 3
**Rating:** 6
**Confidence:** 3

**Summary:**

This paper introduces GeoGramBench, a benchmark to evaluate large language models' ability to perform geometric spatial reasoning from procedural drawing code (e.g., Asymptote), formalizing it as the Program-to-Geometry task. It highlights LLMs' underexplored weaknesses in translating symbolic code into spatial representations. The benchmark comprises 500 curated geometry problems, organized by a novel three-level taxonomy based on geometric complexity (Primitive Recognition, Local Relation Composition, Global Abstract Integration). Drawing from sources like NuminaMath and MATH-500, the dataset addresses challenges like answer leakage through human refinement and decontamination. Evaluations on 17 frontier LLMs are comprehensive and do reveal the persistent deficiencies.

**Strengths:**

- The authors effectively address prevalent issues in similar datasets, such as answer leakage, through meticulous processing techniques that significantly elevate the benchmark's overall quality.
- The evaluation spans 17 open- and closed-source models of varying scales, uncovering both the capabilities and shortcomings of existing LLMs in this domain.
- The paper is generally well-written, with smooth and coherent flow throughout.

**Weaknesses:**

- It would be valuable to incorporate evaluations using vision models that directly interpret the rendered images, alongside results from multimodal models, to serve as a reference baseline. This would help disentangle whether the observed performance gaps arise from deficiencies in reasoning capabilities or challenges in parsing Asymptote code.
- There is a discrepancy between Table 1 and the corresponding text on page 8: "For example, GPT-o1 drops from 76.02% to 43.35%, and DeepSeek-R1 drops from 75.27% to 40.38%." Please verify and ensure alignment between the visual and textual elements.
- The paper would benefit from a standardized definition of the dataset's input-output format, including an accessible, introductory explanation of Asymptote code to enhance clarity for readers unfamiliar with procedural drawing languages.

**Questions:**

- Could the paper include a formal definition of "Asymptote code" to provide clarity for readers? Additionally, it would be helpful to elaborate on the process of standardizing procedural code from diverse data sources into a consistent format.
- Although benchmarks in this niche are scarce, it would strengthen the discussion to explicitly compare the proposed work with related efforts, such as [1].

[1] Qiu Z, Liu W, Feng H, et al. Can large language models understand symbolic graphics programs?

---

> ### Author Response · Authors · 2025-11-23
> **Author Response to Reviewer XpCL (Part 1/2)**
>
> Thank you for your thoughtful review and constructive comments. Below are our responses to your suggestions. We would greatly appreciate any further feedback to help us improve our work.
>
> ---
>
> >**Q1**. It would be valuable to incorporate evaluations using vision models that directly interpret the rendered images, alongside results from multimodal models, to serve as a reference baseline. This would help disentangle whether the observed performance gaps arise from deficiencies in reasoning capabilities or challenges in parsing Asymptote code.
>
> **A1**:
> Thank you for this valuable suggestion regarding the inclusion of vision model baselines to help disentangle whether performance gaps are due to reasoning capabilities or challenges in parsing Asymptote code.
>
> To address this, we conducted additional experiments using three state-of-the-art visual-language models (VLMs): Qwen3-VL-2B-Thinking, Qwen3-VL-4B-Thinking, and Qwen3-VL-8B-Thinking, which have strong reasoning abilities and also support visual inputs. In line with Section 3.2 of our paper, we evaluated these models on the subset of MATH-500 problems that include geometric drawing programs, comparing performance when providing either the rendered geometric image or the drawing code as input, alongside the problem statement.
>
> The results are as follows:
>
> | Input Type                | Qwen3-VL-8B-Thinking | Qwen3-VL-4B-Thinking | Qwen3-VL-2B-Thinking |
> |-------------------------------|----------------------|----------------------|----------------------|
> | Visual Image + Problem        |      66.67          |     59.52           |       54.76         |
> | Drawing Code + Problem        |      73.81          |     64.29           |       57.14         |
>
> We observe that providing the geometric drawing code typically leads to higher accuracy than supplying the visual rendering. This suggests that parsing Asymptote code is not the main challenge for LLMs in our setting; rather, symbolic programmatic representation offers a clearer and more structured way to describe geometry, which models may process more effectively than visual input. This finding is consistent with SGP-Bench (ICLR 2025), which also reports that symbolic representations are generally more precise and less affected by the confounding factors that can impact visual perception. Overall, our results reinforce the value of using symbolic programmatic representations for geometric reasoning in LLMs.
>
> ---
>
> >**Q2**. There is a discrepancy between Table 1 and the corresponding text on page 8: "For example, GPT-o1 drops from 76.02% to 43.35%, and DeepSeek-R1 drops from 75.27% to 40.38%." Please verify and ensure alignment between the visual and textual elements.
>
> **A2**:
> Thank you for drawing our attention to the discrepancy between Table 1 and the text on page 8. We have carefully checked and corrected the inconsistency, and we have also reviewed the entire manuscript to ensure that all data in the tables and text are now fully aligned. We appreciate your careful reading and have addressed this issue in our revised submission.

---

> ### Author Response · Authors · 2025-11-23
> **Author Response to Reviewer XpCL (Part 2/2)**
>
> >**Q3**. The paper would benefit from a standardized definition of the dataset's input-output format, including an accessible, introductory explanation of Asymptote code to enhance clarity for readers unfamiliar with procedural drawing languages.
>
> > Could the paper include a formal definition of "Asymptote code" to provide clarity for readers? Additionally, it would be helpful to elaborate on the process of standardizing procedural code from diverse data sources into a consistent format.
>
>
> **A3**: Thank you for the valuable suggestions concerning the standardized input-output format and the definition and handling of Asymptote code in our dataset. We have incorporated these clarifications and revisions into both the main text and appendix of the updated manuscript.
>
> **1.Standardized Definition of Input-Output Format**
> To enhance clarity, we have added Section 3.3 "TASK DATA FORMAT" to the paper.
> - Each problem in our benchmark consists of two components: a textual description and geometric drawing code, written in a procedural drawing language such as Asymptote or Matplotlib style. The model receives both the text and code as input. The expected output is a numeric answer, which may correspond to length, area, volume, angle, ratio, or count. This format requires the model not only to parse and understand the procedural code, but also to construct an accurate internal geometric representation for reasoning and calculation.
>
> **2. Asymptote Language and Code Standardization**
> We have clarified these points in the revised manuscript and appendix:
>
> - **Introduction to Asymptote Language** Asymptote is a vector graphics language designed for technical drawing and geometric visualization. "Asymptote code" refers to the structured source code written in this language, used to specify geometric primitives (points, lines, circles, polygons, 2D/3D shapes), spatial relationships, and drawing instructions. Its syntax supports mathematical expressions, procedural constructs, and graphical commands for precise, programmatic construction of geometric diagrams.
>
> - **Standardization of Procedural Code:**
>   Our benchmark curation pipeline involves the following steps:
>   1. Initial Filtering: We collect mathematical problems and identify those containing embedded Asymptote code by searching for `[asy]` and `[/asy]` tags.
>   2. Syntax Validation: Using the Asymptote rendering tool, we check the correctness of each code snippet and exclude those with errors.
>   3. Diagram Verification and Adjustment: We ensure each code snippet renders a clear and valid geometric diagram. Canvas size and proportions are adjusted as needed for clarity.
>   4. Answer Leakage Prevention: As described in Section 4.1, we review code segments for both direct and indirect answer leakage. Direct leakage (answers explicitly present in code) is handled by rescaling or modifying the geometry. Indirect leakage (answers inferable from code parameters) is prevented by masking or altering those parameters.
>
> These clarifications have been incorporated in both the main text and appendix of the revised manuscript to help readers unfamiliar with procedural drawing languages and to ensure consistent and reliable benchmarking.
>
>
> >**Q4**. Although benchmarks in this niche are scarce, it would strengthen the discussion to explicitly compare the proposed work with related efforts, such as [1].
>
> **A4**: Thank you for highlighting the importance of comparing our work to related efforts. We have cited SGP-Bench (ICLR 2025) and included a discussion of its relevance in the related work section.
>
> In response, we have closely examined and summarized the key differences between GeoGramBench and SGP-Bench in our revised manuscript. Both works assess symbolic-to-spatial reasoning in LLMs. SGP-Bench focuses on general semantic understanding and invariance of visual program representations (covering SVG and CAD formats), while GeoGramBench is designed to probe mathematical reasoning over procedural geometric code, specifically targeting the step-by-step construction and reasoning with geometric figures. Our benchmark introduces a three-level taxonomy based on geometric complexity and provides detailed failure analysis on how LLMs handle increasingly complex geometry tasks.
>
> Overall, GeoGramBench complements the broader scope of SGP-Bench by offering rigorous evaluation in the mathematical geometry domain and deeper analysis of chain-of-thought reasoning and symbolic figure construction. We have added this comparative discussion to the related work section to clarify the positioning of our contribution and appreciate the suggestion to make this connection explicit.

---

### Official Review · Reviewer_QZFV · 2025-10-31

**Soundness:** 3
**Presentation:** 2
**Contribution:** 2
**Rating:** 4
**Confidence:** 4

**Summary:**

The authors of this paper formalizes this novel task called Program-to-Geometry, which tests the model's abilities make geometric reasoning based on programmatic drawing code. The proposed GeoGramBench, consists of 500 curated problems incorporating programmatic drawing code and evaluated the performance of the 17 current state-of-the-art models on their proposed benchmark.

**Strengths:**

- interesting insights have been provided in the section 6 "behavior analysis of LLMs"
- detailed description of the data curation process (pre-selection, deduplication, human verification and answer leakage prevention)

**Weaknesses:**

- a more detailed analysis for the main results table in Section 5.3 is required, it is interesting that they struggle with angle and volumne, but lacks a bit of insights (for example why is this not the most challenging questions for abstract?)
- I am not entirely sure when the experiments have been drafted, but maybe add one most recent model for evaluation?
- The tasks seems undoubtly interesting, and one could see the implication, but testing the model to make geometric reasoning based on procedural code (albeit might have different syntax, but the geometric properties of these DSL are similar, information about the 2D/3D properties are stored somewhere in the code) has been studied for example in SGPBench (similar to the tested ability b in section 3.1 to make spatial and geometric reasoing) and SGP-GenBench (similar to ability a in section 3.1 generate geometric diagrams) and other similar works, in fact a lot of works has been dedicated for that, I would want to have some discussions with benchmarks from that area in the discussion / related work sections
- in general I am open for discussion and modification of my ratings

**Questions:**

- benchmark is important and interesting for the community to push the boundary of current AI systems, but there are many capabilities that we could that test, the more interesting problems are what are the implications of this benchmark? how can we use this benchmark to improve the current models? if we find a way to improve the models to be better in the proposed benchmarks, are they domain-specific capabilities or can they benefit other reasoning abilities?
- the authors seem to be discovering that the CoT might not be beneficial for this task, which is fine per se, however, since CoT or different variants are standard approaches for improving the reasoning performance, why does it fail on GeoGramBench? might it be an indication that the tested Program-to-Geometry ability to be a very specific and not generic ability?
- what does the level 1-5 in Figure 2 mean?
- I am a bit confused with figure 1: math-500 and aime 24 are text-only benchmarks, where you getting the visualizations? let's say you constructed it, but you are suggesting in this diagram, that by providing additional visualization to the already existing text and code for geometric shapes, the performance actually gets worse?

---

> ### Author Response · Authors · 2025-11-23
> **Author Response to Reviewer QZFV (Part 1/4)**
>
> Thank you for your thoughtful review and constructive comments. In the following, we provide additional clarifications and experimental results based on your feedback. Please let us know if you have any further questions or suggestions.
>
> ---
>
> >**Q1**. a more detailed analysis for the main results table in Section 5.3 is required, it is interesting that they struggle with angle and volumne, but lacks a bit of insights (for example why is this not the most challenging questions for abstract?)
>
> **A1**: Thank you for your suggestion regarding deeper analysis of the main results table in Section 5.3. We have incorporated a more detailed analysis in the revision, summarized as follows:
>
> 1. **Closed-source models outperform open-source models.** The latest GPT-5 model achieves state-of-the-art performance, with an overall average accuracy of 75.01%, and leads on both the Primitive and Compositional levels. Among open-source models, Qwen3-235B-Thinking-2507 ranks second at 74.00%, ahead of GPT-o1 and GPT-o3-mini, which both average around 70%.
>
> 2. **All models perform below 50% in the Abstract level.** As the complexity of geometric programs rises, accuracy drops sharply for all models on Abstract level, with none surpassing 50%. For example, Qwen3-235B-Thinking-2507 achieves 89.09% on Primitive and 79.12% on Compositional, but just 49.05% on the Abstract level. GPT-5 follows a similar trend, dropping to 39.26% on Abstract.
>
> 3. **Most challenging subtypes: angle and volume.**
> - At the Primitive and Compositional levels, which focus on 2D geometric figures, the angle subtype is most challenging, as it requires the model to reconstruct and reason about implicit spatial relationships in 2D, which is difficult with only code input. In contrast, length or count subtypes are typically more straightforward, often solvable with direct algebraic or counting methods.
> - At the Abstract level, dominated by 3D geometric figures, the most difficult subtypes shift to area and especially volume. Area and volume problems require complete 3D spatial understanding, making them especially difficult at the abstract level. This means models struggle much more with these tasks. Angle problems can sometimes be simplified to 2D scenarios using vector or triangle-based techniques, so their relative difficulty is lower compared to area and volume. The key point is not that angle questions become simple, but that area and volume tasks in 3D geometry present a much greater challenge for current models. Thus, the stark accuracy drop for area and volume in the Abstract level reflects the greater demand for true 3D spatial integration and highlights the significant challenges that current models face in higher-dimensional reasoning.
>
>
> ---
>
> >**Q2**. I am not entirely sure when the experiments have been drafted, but maybe add one most recent model for evaluation?
>
> **A2**:
> Thank you for your suggestion to include more recent models in our evaluation. In the revised experiments, we have added the newest closed-source model GPT-5, as well as the latest open source model Qwen3-32B, which is recognized as one of the strongest reasoning models among 32B parameter LLMs.
>
> 1. We observe that the capabilities of state-of-the-art models continue to improve. For example, GPT-5 achieves a new SOTA average accuracy on GeoGramBench, with over 90% accuracy at the Primitive level, indicating that the industry's top models are making steady and significant advances. Similarly, Qwen3-32B achieves state-of-the-art results among models of the same scale, showing that both closed and open-source models are reaching new performance milestones.
>
> 2. However, the main conclusion remains unchanged on GeoGramBench: as the complexity of geometric figures increases, model accuracy drops. Even though GPT-5 demonstrates outstanding performance with 90% accuracy on the Primitive level, its accuracy falls sharply to only 39.26% at the Abstract level. These results suggest that while newer models greatly improve perception and reasoning for basic geometric figures, there is still little progress in handling complex and abstract geometric structures.

---

> ### Author Response · Authors · 2025-11-23
> **Author Response to Reviewer QZFV (Part 2/4)**
>
> >**Q3**. The tasks seems undoubtly interesting, and one could see the implication, but testing the model to make geometric reasoning based on procedural code (albeit might have different syntax, but the geometric properties of these DSL are similar, information about the 2D/3D properties are stored somewhere in the code) has been studied for example in SGPBench (similar to the tested ability b in section 3.1 to make spatial and geometric reasoing) and SGP-GenBench (similar to ability a in section 3.1 generate geometric diagrams) and other similar works, in fact a lot of works has been dedicated for that, I would want to have some discussions with benchmarks from that area in the discussion / related work sections
>
>
> **A3**: Thank you for highlighting the connection to prior work on symbolic graphics and procedural code for geometric reasoning. In the revised manuscript, I have added a section in Related Work to discuss this literature in more depth.
>
> **Symbolic Graphics Perception and Generation**  A growing line of work targets the symbolic understanding and manipulation of SVG code for LLMs. Benchmarks such as SGP-Bench [1], SVG Taxonomy [2], and SVGenius [3] focus on SVG code parsing and perception, probing models’ abilities to interpret, understand, and reason about symbolic vector graphics. Other efforts, such as SGP-GenBench [4], provide benchmarks for evaluating models’ capabilities in SVG code generation, while methods like Chat2SVG [5] propose approaches for improving SVG generation from natural language instructions. Together, these works lay important foundations for assessing both the perception and generation aspects of symbolic graphics in LLMs. However, there is still a lack of benchmarks specifically designed to test the construction of geometric spatial structures and deeper reasoning within geometry, which is the focus of GeoGramBench.
>
>
> The key distinction of GeoGramBench is its focus on deeper geometric reasoning and mathematical computation within geometric spaces, especially the need for models to construct internal representations and solve geometry problems, not just perception or code synthesis. This aspect remains underexplored in existing work. We have included these discussions and references in the revised Related Work section.
>
> [1] Can large language models understand symbolic graphics programs?. ICLR 2025.
> [2] Exploring the capability of llms in performing low-level visual analytic tasks on svg data visualizations. VIS 2024.
> [3] Benchmarking llms in svg understanding, editing and generation. ACMMM 2025.
> [4] Symbolic graphics programming with large language models. Arxiv 2025.
> [5] Chat2svg: Vector graphics generation with large language models and image diffusion models. CVPR 2025.
>
> ---
>
> >**Q4**. the authors seem to be discovering that the CoT might not be beneficial for this task, which is fine per se, however, since CoT or different variants are standard approaches for improving the reasoning performance, why does it fail on GeoGramBench? might it be an indication that the tested Program-to-Geometry ability to be a very specific and not generic ability?
>
> **A4**: Thank you for your insightful question. In our study, we observed that CoT does not significantly improve performance in Program-to-Geometry tasks. This may indicate a new and broader challenge for CoT approaches in spatial reasoning scenarios.
>
> Program-to-Geometry is an important subdomain of spatial reasoning where models need to interpret geometric structures from symbolic code. Our paper demonstrates through quantitative and qualitative experiments that CoT does not reliably improve performance for this type of task. While CoT is effective in many conventional reasoning scenarios, it consistently struggles with spatial or geometric reasoning based on symbolic inputs.
>
>
> This challenge is not unique to Program-to-Geometry. Recent work, such as Qwen3-VL (https://qwen.ai/blog?id=99f0335c4ad9ff6153e517418d48535ab6d8afef&from=research.latest-advancements-list), shows that even in broader vision-based spatial benchmarks, specifically **Embodied and Spatial Understanding**, reasoning models only match or sometimes underperform compared to instruct models.
>
>
>
> In summary, our findings reveal that there are significant challenges for CoT in geometric symbolic spatial reasoning tasks, and this may be a general issue for spatial reasoning, not just specific to Program-to-Geometry.

---

> ### Author Response · Authors · 2025-11-23
> **Author Response to Reviewer QZFV (Part 3/4)**
>
> >**Q5**. benchmark is important and interesting for the community to push the boundary of current AI systems, but there are many capabilities that we could that test, the more interesting problems are what are the implications of this benchmark? how can we use this benchmark to improve the current models? if we find a way to improve the models to be better in the proposed benchmarks, are they domain-specific capabilities or can they benefit other reasoning abilities?
>
> **A5**: Thank you for this valuable question about the implications and broader impact of our work. GeoGramBench has several important implications for model evaluation and future research:
>
> **Symbolic geometric representation is more effective than visual input.**  Experiments in SGP-Bench and our own work show that using symbolic procedural code to describe geometry leads to higher model accuracy compared to visual images. For example, when evaluating state-of-the-art visual-language models (Qwen3-VL-2B/4B/8B) on the subset of MATH-500 problems with geometric drawing code, we find that providing the symbolic code results in better accuracy than supplying rendered images. The results below illustrate this finding:
>
> | Input Modality                | Qwen3-VL-8B-Thinking | Qwen3-VL-4B-Thinking | Qwen3-VL-2B-Thinking |
> |-------------------------------|----------------------|----------------------|----------------------|
> | Visual Image + Problem        |      66.67           |     59.52            |       54.76          |
> | Drawing Code + Problem        |      73.81           |     64.29            |       57.14          |
>
>
> This outcome suggests that describing geometric figures with symbolic procedural code is both more efficient and more accurate than using visual images. Symbolic representations provide structured information that makes it easier for LLMs to construct and reason about geometric relationships.
>
> **Limitations of CoT in symbolic spatial reasoning.**  Our quantitative and qualitative analyses reveal that CoT prompting provides little benefit in symbolic spatial reasoning tasks. This is consistent with findings in vision-based benchmarks, such as those with Qwen3-VL(https://qwen.ai/blog?id=99f0335c4ad9ff6153e517418d48535ab6d8afef&from=research.latest-advancements-list), where reasoning models do not outperform instruct-tuned models in spatial reasoning scenarios.
>
> **Benchmark impact and future promise.**  GeoGramBench is the first formal benchmark to define and comprehensively evaluate the Program-to-Geometry task. This task assesses both the model's ability to construct geometric spaces from symbolic procedural code and its capacity to perform reasoning within those spaces. It contains 500 diverse problems spanning three geometric complexity levels and multiple subtypes (length, angle, area, volume, etc.), allowing for broad and rigorous assessment of LLMs' geometric construction and spatial reasoning abilities. The benchmark helps to expose model weaknesses, especially in handling abstraction and spatial logic, and provides a valuable testbed for developing improved reasoning methods.
>
> Overall, improvements made on this benchmark have the potential to enhance model performance not only within geometric domains but also within broader reasoning tasks that require structured spatial understanding and complex abstraction.

---

> ### Author Response · Authors · 2025-11-23
> **Author Response to Reviewer QZFV (Part 4/4)**
>
> >**Q6**. what does the level 1-5 in Figure 2 mean?
>
> **A6**: Thank you for your question about the meaning of level 1-5 in Figure 2. These levels are the official difficulty labels in the MATH-500 dataset (https://huggingface.co/datasets/HuggingFaceH4/MATH-500), where each problem is rated from 1 (easiest) to 5 (hardest). The complexity level is determined based on factors such as the number of computational or reasoning steps required and the typical grade or curriculum level needed to solve the problem.
>
> In Figure 2, we show model accuracy based on reasoning level for two kinds of questions in MATH-500: those with procedural geometric code and those without. For the subset of questions that include geometric drawing code, models must correctly interpret the geometric figure described by the code to solve the problem. Our analysis shows that, for these questions, accuracy does not decrease simply with higher reasoning level, but is instead much more sensitive to the complexity of the geometric figure itself. As geometric complexity increases, accuracy drops sharply. This highlights that understanding and reasoning over programmatically described geometric figures is a critical challenge for current models, beyond what is captured by reasoning level alone.
>
>
>
> ---
>
> >**Q7**. I am a bit confused with figure 1: math-500 and aime 24 are text-only benchmarks, where you getting the visualizations? let's say you constructed it, but you are suggesting in this diagram, that by providing additional visualization to the already existing text and code for geometric shapes, the performance actually gets worse?
>
> **A7**: Thank you for your question and for pointing out the possible confusion with Figure 1. Both MATH-500 and AIME24 benchmarks contain two types of problems: some are described purely by text (such as algebra questions), while others are described by text along with drawing code (in Asymptote), typically used for geometry problems. The latter category specifically examines the ability of LLMs to understand and reason about geometric figures through procedural code.
>
> In Figure 1(a), the visualized geometric figure is constructed by extracting an example from the MATH-500 dataset and rendering it using the Asymptote drawing tool. This visualization is provided for the reader’s understanding only and is **not** given as input to the LLM.
>
>
> Figures 1(b) and 1\(c) compare model accuracy on text-only problems with model accuracy on problems that include both textual descriptions and drawing code within MATH-500 and AIME24. Our results show that for problems with text plus drawing code, model accuracy drops noticeably compared to text-only problems. This indicates that models face significant challenges in building correct spatial representations from symbolic code, which was a key motivation for the further research presented in our work.

---

### Official Review · Reviewer_btiM · 2025-11-01

**Soundness:** 3
**Presentation:** 3
**Contribution:** 2
**Rating:** 6
**Confidence:** 4

**Summary:**

The paper formalizes Program‑to‑Geometry: reading procedural drawing code (Asymptote/Matplotlib‑style) and performing downstream geometric reasoning. It releases GeoGramBench, 500 curated problems with a three‑level taxonomy 1)Primitive Recognition, 2) Local Relation Composition, and 3) Global Abstract Integration, organized by geometric complexity rather than typical “reasoning complexity,” plus six subtypes from "Angle", to "Count". Across 17 models, accuracy drops sharply at the highest abstraction level, with no model exceeding ~50% there.

**Strengths:**

1. Code-based spatial reasoning is an interesting and under-explored domain.
2. Clear task definition and a taxonomy grounded in geometric (not step-count) complexity; the benchmark isolates program-to-diagram understanding from general math difficulty.
3. Broad evaluation over 17 models with consistent prompting and multi-sample reporting; the “<50% on Abstract” result is a useful negative finding that signals a genuine capability gap.
4. The ANSWER LEAKAGE CHALLENGES is sound and interesting.

**Weaknesses:**

1. Positioning vs. prior art (major). The paper does not cite SGP-Bench (ICLR 2025), which evaluates LLMs’ understanding of symbolic graphics programs without rendering as well, at broader scope (SVG for 2D; CAD for 2D/3D) and with systematic invariance tests under program transformations. While GeoGram focuses more on the 2D geometry math aspect, I think SGP-Bench is closely related and predates this submission, it should be discussed and contrasted.
2. GeoGramBench stays within Asymptote/Matplotlib-style code and contest-style numeric targets. That makes results crisp, but limits to the mathematical geometry domain which is a fraction within the broad discussion of geometric spatial capability.
3. While the paper's goal is to evaluate the program-to-geometry / geo-spatial capability of the LLMs, the questions also involve mathematic reasoning capability to solve the planar geometry tasks. Therefore, the final accuracy is kind of inconclusive for the geo-spatial one itself.
4. I think the DSL choice (Asymptote/Matplotlib‑style) can significant affect the evaluation results since various LLMs could be exposed to different amount of DSL code training. few-shot CoT experiments are important to justify the evaluation.

**Questions:**

I would like know the authors' thoughts and opinions regarding the weaknesses section.

---

> ### Author Response · Authors · 2025-11-23
> **Author Response to Reviewer btiM (Part 1/3)**
>
> We would like to thank the reviewer for providing thoughtful and constructive criticism. We have taken all comments seriously and have done our best to address each concern in our response. Please let us know if there are any further improvements we can make.
>
> ---
>
> >**Q1**. "Positioning vs. prior art (major). The paper does not cite SGP-Bench (ICLR 2025), which evaluates LLMs’ understanding of symbolic graphics programs without rendering as well, at broader scope (SVG for 2D; CAD for 2D/3D) and with systematic invariance tests under program transformations. While GeoGram focuses more on the 2D geometry math aspect, I think SGP-Bench is closely related and predates this submission, it should be discussed and contrasted."
>
>
> **A1**: Thank you for this constructive suggestion regarding the citation and positioning with respect to SGP-Bench (ICLR 2025). We agree that SGP-Bench is highly relevant to our work, as it also investigates the capability of LLMs to understand symbolic spatial representations. Based on your advice, we have cited SGP-Bench and added a detailed comparison to the Related Works section of our revised manuscript.
>
> Upon close examination, we identify several key differences between GeoGramBench and SGP-Bench.
> -  **Motivation:** While both benchmarks address symbolic-to-spatial reasoning in LLMs, SGP-Bench primarily focuses on general semantic understanding (such as shape, color, and count) and on invariance under program transformations, whereas GeoGramBench emphasizes geometric spatial abstraction and requires models to solve mathematical problems based on geometric understanding, including the calculation of angles, areas, and volumes.
> - **Methodology:** GeoGramBench introduces a three-level taxonomy reflecting geometric complexity, which allows for fine-grained assessment of model capabilities ranging from primitive element recognition to local relation composition and global abstract integration of figures. We also use numeric-answer questions, which are better suited for evaluating mathematical reasoning abilities.
> - **Conclusions:** Our study finds that while top models achieve over 80% accuracy on primitive geometric tasks, their performance drops below 50% as diagram complexity increases. We provide a qualitative failure analysis, observing that models often prefer coordinate-based calculations and rarely use auxiliary geometric constructions, even when such methods would be more effective.
>
> In summary, while our work and SGP-Bench both evaluate LLMs on symbolic-to-spatial reasoning, our benchmark addresses distinct methodological questions and provides complementary insights into the limitations of models on geometry math tasks. SGP-Bench is now cited and discussed in the related work section of our revised manuscript.
>
> ---
>
> >**Q2**. GeoGramBench stays within Asymptote/Matplotlib-style code and contest-style numeric targets. That makes results crisp, but limits to the mathematical geometry domain which is a fraction within the broad discussion of geometric spatial capability.
>
>
> **A2**:  Thank you for raising this important point about the scope and design choices of GeoGramBench. We agree that these aspects are central to how well our benchmark can reflect the broader capabilities of geometric spatial reasoning.
>
> - Regarding the choice of drawing code language, we acknowledge that using a wider range of languages might further broaden the evaluation. However, as demonstrated in our ablation experiments (see Appendix A), the selection between Asymptote and Matplotlib-style code does not significantly impact model performance. This conclusion is consistent with SGP-Bench (ICLR 2025), which also found the main challenge is not code syntax, but rather the model's ability to construct and reason over complex spatial structures from symbolic input. For this reason, we mainly use Asymptote in our benchmark.
>
> - With respect to the use of numeric targets, we deliberately employ geometry problems requiring numeric answers to more accurately assess the model's reasoning process. Numeric answers, as opposed to multiple-choice or proof-based formats, minimize ambiguity and guessing, and require the model to carry out explicit, step-by-step reasoning. This format is widely accepted in the community and allows us to clearly track and analyze model errors as well as reasoning chains. As discussed in Section 6, this setup helps us to identify and understand unique model failure patterns that are quite different from those observed in humans.
>
> In summary, while GeoGramBench is focused on the mathematical geometry domain, this controlled and precise setting enables clear, reproducible, and rigorous evaluation of symbolic-to-geometric reasoning.

---

> ### Author Response · Authors · 2025-11-23
> **Author Response to Reviewer btiM (Part 2/3)**
>
> >**Q3**. While the paper's goal is to evaluate the program-to-geometry / geo-spatial capability of the LLMs, the questions also involve mathematic reasoning capability to solve the planar geometry tasks. Therefore, the final accuracy is kind of inconclusive for the geo-spatial one itself.
>
> **A3**: Thank you for highlighting the distinction between program-to-geometry ability and mathematical reasoning. Our preliminary experiments and benchmark analysis show that model accuracy is mainly determined by program-to-geometry capability, rather than mathematical reasoning difficulty.
>
>
> - **Mathematical Reasoning Ability:**  As discussed in Section 1 and Section 3 of our paper, we conducted preliminary experiments comparing LLM performance on two benchmarks, AIME24 and MATH-500, each containing a subset of problems with embedded geometric drawing code. We observed that DeepSeek-R1 showed a drop in accuracy of 23.5% on the code-containing problems in AIME24 and 10.9% in MATH-500, compared to their respective text-only subsets. Importantly, our further analysis found that these drops in accuracy are less related to traditional reasoning complexity and are primarily tied to the presence and intricacy of geometric code. This suggests that the symbolic-to-spatial translation required for interpreting geometric code is a significant challenge for LLMs, beyond the underlying mathematical reasoning itself.
>
> - **Program-to-Geometry Ability:**  To further investigate the challenge of symbolic code understanding and spatial reasoning, we formally defined the program-to-geometry task and created the GeoGramBench benchmark. Problems in GeoGramBench are divided into three levels based on geometric complexity. Our experiments show a clear trend: as the complexity of geometric diagrams increases, model accuracy drops sharply. For example, GPT-o1 achieves 85.92% accuracy at the Primitive level and 76.12% at the Compositional level, but the score drops to just 44.67% at the Abstract level. This result highlights that with more challenging geometric structures, current models increasingly struggle to understand and reason over symbolic code, revealing significant gaps in their spatial reasoning ability.
>
> In summary, while it is true that mathematical reasoning is a part of the task, our experiments and analysis are specifically designed to isolate and assess the symbolic geometric perception and spatial reasoning of LLMs.

---

> ### Author Response · Authors · 2025-11-23
> **Author Response to Reviewer btiM (Part 3/3)**
>
> >**Q4**. I think the DSL choice (Asymptote/Matplotlib‑style) can significant affect the evaluation results since various LLMs could be exposed to different amount of DSL code training. few-shot CoT experiments are important to justify the evaluation.
>
>
> **A4**: Thank you for raising the important question regarding the impact of DSL choice and the role of few-shot CoT experiments in our evaluation. To address this, we conducted experiments in two directions: the influence of different DSLs, and the effect of few-shot CoT prompting.
>
> For DSL choice, we performed ablation studies comparing model performance on the same set of problems written in both Asymptote and Matplotlib-style code. The results (see Appendix A) show only minimal differences, demonstrating that familiarity with the specific DSL has little impact. This suggests that models are generally capable of parsing different code formats, and that the main challenge lies in constructing the underlying geometric spatial structure from the code rather than code parsing itself.
>
> For few-shot CoT experiments, we evaluated whether adding few-shot reasoning examples improves performance. Surprisingly, we found that few-shot CoT did not help and often even reduced accuracy on reasoning tasks, consistent with recent literature [1,2,3]. This result held true especially for reasoning-focused models, where additional examples did not yield improvement.
>
> | Model                    | Few shot | Primitive | Compositional | Abstract | ALL   |
> |:-------------------------|:--------:|:---------:|:-------------:|:--------:|:------|
> | DeepSeek-Distill-Qwen-32B | 0        | 79.78     | 67.83         | 35.92    | 62.68 |
> | DeepSeek-Distill-Qwen-32B   | 1        | 65.20     | 56.41         | 27.84    | 51.40 |
> | DeepSeek-Distill-Qwen-32B| 3        | 56.37     | 40.99         | 18.59    | 38.80 |
> | QwQ-32B                  | 0        | 85.17     | 73.12         | 37.92    | 67.20 |
> | QwQ-32B                  | 1        | 75.98     | 64.25         | 35.29    | 59.75 |
> | QwQ-32B                  | 3        | 63.97     | 47.17         | 19.75    | 44.13 |
> | s1.1-32B                  | 0        | 75.37     | 58.96         | 26.58    | 54.60 |
> | s1.1-32B                  | 1        | 50.49     | 38.53         | 17.23    | 35.90 |
> | s1.1-32B                  | 3        | 34.31     | 21.68         | 8.82     | 21.20 |
>
> In summary, we have directly addressed both the potential effect of DSL choice and few-shot CoT experiments in our work. Our tests show that few-shot CoT does not help and can even reduce performance for reasoning models, and that the influence of DSL code choice on overall evaluation is minimal.
>
> [1] Revisiting Chain-of-Thought Prompting: Zero-shot Can Be Stronger than Few-shot. Arxiv 2025
> [2] Prompt Engineering with Reasoning Models. <https://www.prompthub.us/blog/prompt-engineering-with-reasoning-models>
> [3] Understanding Reasoning LLMs. <https://magazine.sebastianraschka.com/p/understanding-reasoning-llms>

---

### Author Response · Authors · 2025-11-23
**Revision Summary**

Dear Reviewers and ACs,

We sincerely thank all reviewers for their constructive feedback and suggestions. We have carefully revised the manuscript, with changes highlighted in blue. Below, we summarize our main revisions and how we have addressed the key concerns:

1. Related Symbolic-to-Spatial Work:  In response to reviewer suggestions, the Related Work section has been expanded with a dedicated “Symbolic Graphics Perception” paragraph. This now discusses SGP-Bench (ICLR 2025) and other relevant benchmarks, making our connections and distinctions to prior work more explicit.

2. Clarification of GeoGramBench’s Scope and Task Design:  To address points raised by reviewers btiM and Dim1 regarding the balance of mathematical and geometric reasoning, as well as input-output clarity, we have added Section 3.3. This section clearly defines the standardized input-output formats. An introductory explanation of Asymptote code is included in Appendix C.1 for readers unfamiliar with procedural drawing languages, and Appendix C.2 details the process of standardizing code from different sources.

3. Enhanced Empirical Analysis and Experimental Details:  Following reviewer QZFV’s feedback, we have rewritten and deepened the analysis in Section 5.3 to provide more insight into model performance across various geometry categories (e.g., struggles with angles and volumes). We have also included evaluations of the latest models in Section 5.2. To address reviewer Dim1's concern, we now report standard deviations in Appendix F, providing a quantification of performance variability for greater statistical rigor.

4. Benchmark Impact and Implications: GeoGramBench is the first formal benchmark to systematically define and evaluate the Program-to-Geometry task, probing LLMs’ abilities in geometric construction and spatial reasoning. With 500 problems across varying complexity and types, it exposes important model limitations and serves as a practical platform for investigating models' ability to construct geometric spaces from symbolic procedural code and perform reasoning within those spaces.

We have revised the manuscript according to the reviewers’ comments, and we appreciate your thoughtful suggestions. Please see the updated version, and feel free to let us know if there are any remaining questions or feedback.

Best,

Authors

---

### Meta-Review · Area_Chair_CFCy · 2025-12-20

**Summary:**

The reviewers pointed out several weaknesses and shared common concerns, including (a) lack of discussion and comparison with relevant papers; (b) limited scope to the mathematical geometry domain instead of general spatial capability; (c) the mathematical reasoning required may limit evaluation of general geo-spatial capability as intended; (d) lack of in-depth analysis for the failure modes; (e) lack of recent models in the evaluation.

**Reviewer Concerns:**

Reviewer btiM concerns seem to be partly addressed. The concern related to measuring the geo-spatial capabilities alone seems to be outstanding.

Reviewer QZFV concerns seem to be addressed.

Reviewer XpCL concerns seem to be addressed.

Reviewer Dim1 concerns seem to be addressed.

**Reviewer Scores:**

Reviewer btiM would have likely kept their score at 6.

Reviewer QZFV would have likely increased their score from 4 to 6.

Reviewer XpCL would have likely kept their score at 6.

Reviewer Dim1 would have likely increased their score from 6 to 8 (as per the discussion comments).

---

### Decision · Program_Chairs · 2026-01-26

Accept (Poster)